# The Effect of Lifestyle Interventions on Anxiety, Depression and Stress: A Systematic Review and Meta-Analysis of Randomized Clinical Trials

**DOI:** 10.3390/healthcare12222263

**Published:** 2024-11-13

**Authors:** Sohrab Amiri, Nailah Mahmood, Syed Fahad Javaid, Moien AB Khan

**Affiliations:** 1Spiritual Health Research Center, Lifestyle Institute, Baqiyatallah University of Medical Sciences, Tehran 17166, Iran; rsr.amiri.s@bmsu.ac.ir; 2Division of Health Research, Lancaster University, Lancaster LA1 4YW, UK; n.mahmood2@lancaster.ac.uk; 3Health and Wellness Research Group, Department of Psychiatry and Behavioral Sciences, College of Medicine and Health Sciences, United Arab Emirates University, Al-Ain 15551, United Arab Emirates; 4Health and Wellness Research Group, Department of Family Medicine, College of Medicine and Health Sciences, United Arab Emirates University, Al-Ain 15551, United Arab Emirates

**Keywords:** lifestyle, anxiety, depression, stress, mental disorders, systematic review, meta-analysis

## Abstract

Background/Objectives: Depression, anxiety, and stress are common mental health issues that affect individuals worldwide. This systematic review and meta-analysis examined the effectiveness of various lifestyle interventions including physical activity, dietary changes, and sleep hygiene in reducing the symptoms of depression, anxiety, and stress. Using stress as an outcome and conducting detailed subgroup analyses, this study provides novel insights into the differential effects of lifestyle interventions across diverse populations. Methods: Five databases were systematically searched: PubMed, Web of Science, Scopus, Cochrane Library, and Google Scholar, for gray literature searches. Keywords were used to search each database. The search period was from the conception of the databases until August 2023 and was conducted in English. For each analysis, Hedges’ g was reported with a 95% confidence interval (CI) based on the random-effects method. Subgroups were analyzed and heterogeneity and publication bias were examined. Results: Ninety-six randomized clinical trial studies were included in this meta-analysis. Lifestyle interventions reduced depression (Hedges g −0.21, 95% confidence interval −0.26, −0.15; *p* < 0.001; *I*^2^ = 56.57), anxiety (Hedges g −0.24, 95% confidence interval −0.32, −0.15; *p* < 0.001; *I*^2^ = 59.25), and stress (−0.34, −0.11; *p* < 0.001; *I*^2^ = 61.40). Conclusions: Lifestyle interventions offer a more accessible and cost-effective alternative to traditional treatments and provide targeted benefits for different psychological symptoms.

## 1. Introduction

Mental disorders are among the conditions that place a high burden on healthcare [1] and remain among the primary causes of disease burden worldwide. However, there is no evidence of a decrease in this burden compared to previous decades [2]. Depression and anxiety are two categories of common mental disorders with a significant health burden [2]. In 2021, the Global Burden of Disease Study indicated that depression and anxiety are leading causes of disability, and both are among the 25 leading causes of disability worldwide [2,3]. The global prevalence of depressive disorders in 2019 was equal to 3440.1 per 100,000, and the prevalence rates for men and women were 2713.3 and 4158.4 per 100,000, respectively [2]. In addition, the prevalence of anxiety disorders is 3379.5/100,000, and the prevalence rates for men and women are 2859.8 and 4694.75 per 100,000, respectively [2]. The COVID-19 pandemic has affected public health as well as social structures, contributing to a significant increase in stress levels and further exacerbating mental health challenges worldwide [4,5].

According to a World Health Organization (WHO) report published in 2022, one out of every eight people worldwide live with a mental illness [6]. Thus, globally, 970 million people live with a mental disorder, of which depression and anxiety are the most common [7]. Several factors are associated with the prevalence of depression, anxiety disorders, and stress, including personality traits [8], financial status [9], biological factors [10], parental factors [11], and chronic diseases [12]. Additionally, a class of factors that can affect depression, anxiety, and stress is related to lifestyle, including physical activity [13,14,15,16,17], dietary patterns [18,19], sleep problems [20,21], smoking [22,23], and body mass index [24,25,26].

Lifestyle refers to “the characteristics of inhabitants of a region in special time and place” [27]. Interventions based on a healthy lifestyle have improved physical and mental health, with several studies exhibiting their effectiveness in cases of type 2 diabetes [28,29], obesity [30,31], cardiovascular risk [32], reducing cancer risk [33], obstructive sleep apnea [34], preventing weight gain [35], and mental health [36]. Considering the role of lifestyle interventions in improving health status, studies have investigated their effects in improving mental health and issues related to it [37,38].

A healthy lifestyle effectively reduces depression and anxiety [39,40]. A comprehensive review of studies conducted on the effectiveness of lifestyle interventions for common mental disorders, including depression and anxiety, indicated that there is extensive research available on this field, based on which, review and meta-analysis studies have also been conducted [39,40,41,42,43,44,45,46]. Specifically, for depression, results indicate that Cohen’s effect size ranges from −0.18 to −0.95 [40,41,43] while, for anxiety symptoms, Cohen’s effect size has been reported at −0.19 [39]. Although these reviews and meta-analyses offer valuable insights, they have revealed critical gaps. Most studies have focused on depression and anxiety; however, psychological stress, which is a distinct mental health condition, has not been thoroughly explored [47]. Although the effectiveness of lifestyle interventions may vary across different patient populations, few studies have analyzed the outcomes based on these differences. Given that depression and anxiety are more prevalent among women, the specific impact of lifestyle interventions on these disorders in women has been understudied [48]. Furthermore, various scales have been used to measure depression, anxiety, and stress; however, the potential impact of these differences in measurement tools on study outcomes has not been sufficiently addressed in previous meta-analyses. Recognizing these gaps, this systematic review and meta-analysis aimed to evaluate the effects of lifestyle interventions on depression, anxiety, and stress. Additional objectives include analyzing the influence of these interventions across different population subgroups, specifically among women, and investigating how measurement scales may affect the results.

## 2. Methods

### 2.1. Inclusion and Exclusion Criteria

The population studied in this research were under lifestyle interventions.Eligible studies must have a lifestyle intervention group and a control group.The outcomes examined in these studies were depression, anxiety, and stress.Randomized clinical trials were eligible. Non-randomized clinical trials, quasi-experimental, and cluster randomized clinical trials were not eligible. Quasi-experimental studies were excluded because of the likelihood of confounding variables affecting the internal validity. Furthermore, studies presenting mixed or combined results were omitted to maintain uniformity in the outcome metrics for depression, anxiety, and stress. It was not possible to calculate the exact effect size, because the studies did not report the number of clusters, intra-class correlation was not reported in cluster randomized control trials, and it was not possible to estimate the effect size correctly [49,50], nor pre-post designs.Studies that studied mixed multiple outcomes were not eligible.For some studies. several reports were published and, in these cases, only the study with the best quality was included in the meta-analysis, excluding the rest.

### 2.2. Information Sources

Five databases were systematically searched to retrieve eligible articles: PubMed, Web of Science, Scopus, the Cochrane Library, and Google Scholar. Google Scholar was specifically used to identify gray literature. A set of keywords were used to search each database. In addition, all references for previous review studies in this regard were also searched by one of the authors. The search period was demarcated from the beginning of database formation. The search was conducted in English until August 2023.

### 2.3. Search Strategy

The review protocol has been registered with PROSPERO under the identifier CRD42023390131. Since the data used in this review were gathered from publicly accessible databases and online searches, ethics committee approval and informed consent were not required necessary. The study selection process is illustrated in Figure 1. A syntax of keywords is shown in Appendix A.

### 2.4. Selection Process

Eligible studies were screened and selected as follows. First, each author screened the collection of studies compiled from five databases based on the inclusion and exclusion criteria. Then, the work was divided into eligible articles, where each author reviewed a set of articles. This process was performed independently. Disagreements were resolved by discussing the final articles.

### 2.5. Data Collection Process

All eligible articles were divided among the authors so that each of them could extract the necessary data. After extracting data from each study, all extracted data were checked again. If the study data were insufficient, one of the authors contacted the other to obtain the necessary information.

### 2.6. Data Items

The intervention variable used in the present study consisted of interventions based on lifestyle. For an intervention to be considered as “lifestyle” oriented, at least two components from, the total lifestyle items had to be included. The outcomes of the study were depression, anxiety, and stress. Depression and anxiety are common mental health disorders. Instruments to measure depression, anxiety, and stress were used in this study. All scales used are listed in Table 1.

### 2.7. Study Risk of Bias Assessment

Following PRISMA guidelines, we used the Cochrane Collaboration’s risk of bias tool to evaluate the quality of the included studies [51]. The tool includes five dimensions of quality assessment: selection bias, performance bias, detection bias, attrition bias, and reporting bias. Bias was evaluated by judging each element from the five key domains. Each element was classified as having high, low, or unclear risk of bias. In this qualitative evaluation, the authors entered independently, and then qualitative evaluations were integrated through a discussion of disagreements. Overall, the quality was sufficient to support robust conclusions, with most studies meeting acceptable quality standards. A detailed summary of the risk of bias for each study is provided in Table 1, which aids in interpreting the reliability and generalizability of the meta-analysis findings.

### 2.8. Effect Measures

The effect size used in this study was the standardized mean difference, which was reported in the form of Hedges’ g effect size and 95% confidence interval (CI). Means, standard deviations, and sample sizes were used for each intervention and control group. In cases where these statistics were not reported, the sample size and *p* value were used.

### 2.9. Synthesis Methods

To calculate the effect size, the mean, standard deviation, and sample size of the intervention and treatment groups were extracted in the post-test. In some studies, instead of standard deviation, standard error or confidence interval was reported, and the Cochrane Handbook procedures were used to convert these into standard deviations [52]. Some studies used the mean change or mean difference or other tests to check for differences between the intervention and control groups. In this case, existing procedures were used to calculate the effect size, which included the use of sample size, *p* value, and direction, the details of which are mentioned in the guide [53]. Some studies have reported multiple dependent outcomes, which were transformed using existing procedures using Comprehensive meta-analysis-Version 3.3 software [53,54]. The effect size used in this study was Hedges’ g, and the 95% confidence interval was classified as follows: 0.20 (low), −0.50 (medium), 0.80 (large) [55]. For each analysis, Hedges’ g was reported with a 95% confidence interval (CI) based on the random-effects method. Hedges’ g was used because it considers the sample size, and the studies included in this meta-analysis had different sample sizes [56]. Based on the goals of this study, the following analyses were conducted. The effects of lifestyle interventions on depression, anxiety, and stress were analyzed separately. For each of these analyses, several subgroups were created based on the type of population and scale used to measure depression, anxiety, stress, and sex. Heterogeneity in the studies included in the meta-analysis and publication bias were also examined. These tests were used for the heterogeneity Q test and *I*^2^ [57,58]. An interpretation of *I*^2^ is as follows: may not be important, moderate, substantial, and considerable [59]. For publication bias, these tests used funnel plots [60,61], Egger’s test [62,63], and Trim and fill [64]. Comprehensive meta-analysis-3 software was used in this study [53].

## 3. Results and Discussion

### 3.1. Screened Studies

The studies were screened based on the flowchart shown in Figure 1. After identifying duplicate studies, ineligible studies and other studies that did not meet the inclusion criteria were excluded. Finally, 97 clinical trial studies [65,66,67,68,69,70,71,72,73,74,75,76,77,78,79,80,81,82,83,84,85,86,87,88,89,90,91,92,93,94,95,96,97,98,99,100,101,102,103,104,105,106,107,108,109,110,111,112,113,114,115,116,117,118,119,120,121,122,123,124,125,126,127,128,129,130,131,132,133,134,135,136,137,138,139,140,141,142,143,144,145,146,147,148,149,150,151,152,153,154,155,156,157,158,159,160] were included in this meta-analysis.
healthcare-12-02263-t001_Table 1Table 1Studies included in meta-analysis.Author and YearCountryFollow-UpPopulationAge Sex% Women Sample Size Lifestyle Intervention DefinitionMental Disorders Mental Disorders Scoring Measure Quality Dimensions ResultsN (Mean, Standard Deviation)Selection BiasPerformance BiasDetection BiasAttrition BiasReporting BiasRandom SequenceGenerationAllocation ConcealmentAnderson 2015 [66]Australia3-month Breast cancer45–60Women 51Pink Women’s Wellness Program1-Depression 2-AnxietyHigher scorer indicating more mental problemss 1-GreeneClimacteric Scale Low UnclearHigh High Low Unclear Depression Intervention26 (4.1 ± 2.1)Control25 (4.3 ± 3.6)Anxiety Intervention26 (4.9 ± 3.5)Control25 (4.5 ± 2.8)Azami 2018 [67]Iran 3-month6-month Type2 Diabetes≥1865.5% women 142Nurse-led diabetes self-management1-Depression Higher scorer indicating more mental problemss1-Center for Epidemiologic Studies Depression ScaleLow Low High Low Low Unclear3-monthIntervention71 (11.98 ± 4.97)Control71 (12.84 ± 4.61)6-monthIntervention71 (11.95 ± 5.03)Control71 (12.91 ± 4.5)Brennan 2012 [68]Australia6-month Overweight/Obese11–1954% women 63Cognitive Behavioural Lifestyle1-Depression 2-Anxiety 3-StressHigher scorer indicating more mental problemss1-Depression anxiety and stress scaleLow UnclearUnclearLow Low UnclearDepression Intervention40 (7.9 ± 9.6)Control21 (4.2 ± 5.9)AnxietyIntervention40 (7.0 ± 7.4)Control21 (6.8 ± 6.0)StressIntervention40 (9.9 ± 10.2)Control21 (7.7 ± 7.3)Bringmann 2022 [69]Germany1-month2-month6-month Mild to moderate depression49.1 ± 11.1 in intervention51.0 ± 12.7 in control 77.8% women 54Meditation-Based Lifestyle Modification1-Depression 2-StressHigher scorer indicating more mental problemss1-Beck Depression Inventory 2-Perceived Stress Scale-10Low Low Low Low Low UnclearDepression 1-monthIntervention27 (16.81 ± 10.65)Control27 (20.89 ± 8.14)2-month Intervention27 (13.59 ± 10.63)Control27 (21.59 ± 9.67)6-monthDepression Intervention27 (13.68 ± 10.36)Control27 (20.80 ± 10.95)Stress2-monthIntervention27 (20.11 ± 5.34)Control27 (27.15 ± 4.58)6-monthIntervention27 (20.54 ± 5.64)Control27 (25.85 ± 6.59)Brown 2001 [70]USA8-weekMild to moderate depression19–78Women 104Multi-Modal Intervention1-Depression Higher scorer indicating more mental problems1-Center for Epidemiologic Studies Depression ScaleHigh Low UnclearUnclearLow UnclearDepression Intervention53 (10.4 ± 7.3)Control51 (16.7 ± 10.4)Casañas 2012 [72]Spain 3-month6-month9-month Major depression≥2089.2% women 231Psycho-educational1-DepressionHigher scorer indicating more mental problems1-Beck Depression Inventory Low Low High High Low Unclear3-monthIntervention119 (15.42 ± 7.53)Control112 (17.54 ± 7.18)6-month119 (15.37 ± 8.74)Control112 (16.51 ± 7.60)9-month119 (15.09 ± 8.62)Control112 (16.35 ± 7.84)Cezaretto 2012 [73]Brazil9-month Type 2diabetes18–7967.8% women 177Intensive interdisciplinaryintervention1-DepressionHigher scorer indicating more mental problems1-Beck Depression Inventory UnclearUnclearUnclearUnclearHigh UnclearIntervention75 (8.4 ± 7.7)Control 60 (5.2 ± 5.1)Chang 2018 [74]Korea3-month Older Adults with major depressive disorder77.8 ± 6.687.1% women 93Multi-Domain Lifestyle Modification1-DepressionHigher scorer indicating more mental problems1-GeriatricDepression Scale (GDS)-Short FormLow Low UnclearLow Low UnclearIntervention47 (7.5 ± 4.1)Control46 (10.2 ± 3.6)Charandabi 2017 [75]Iran 2-month spouses of pregnant women31.9 ± 5.3Men 126Life Style Based Education1-Depression2-Anxiety Higher scorer indicating more mental problems1-Edinburgh Postnatal Depression Scale2-Spielberger’s State-Trait Anxiety InventoryLow Low High Low Low UnclearDepression Intervention62 (2.7 ± 3.4)Control63 (4.3 ± 3.8)State anxiety Intervention62 (30.1 ± 7.7)Control63 (35.8 ± 10.5)Trait anxietyIntervention62 (30.7 ± 7.6)Control63 (35.8 ± 9.7)Chiang 2019 [76]Taiwan3-monthMetabolic Syndrome≥40Women 68Lifestylemodification combined with motivational counseling1-DepressionHigher scorer indicating more mental problems1-Beck Depression Inventory Low Low Unclear Low UnclearUnclearIntervention34 (3.8 ± 1.5)Control34 (9.1 ± 6.9)Clark 2012 [77]USA6-month Older people60–9565.9% women 360Lifestyle intervention (Well Elderly Lifestyle)1-Depression Higher scorer indicating more mental problems1-Center for Epidemiologic Studies Depression ScaleLow UnclearLow Low Low UnclearIntervention186 (12.47 ± 9.68)Control173 (13.53 ± 11.17)Croker 2012 [78]UK6-month Obese 10.3 ± 1.669.4% women 63family-based behavioral treatment1-DepressionHigher scorer indicating more mental problems1-Children’s Depression InventoryLow High High Low UnclearUnclearIntervention33 (49.24 ± 6.91)Control30 (48.13 ± 6.97)Desplan 2014 [79]France 1-month obstructive sleep apnea35–70Unknown 22lifestyle intervention1-Depression 2-Anxiety Higher scorer indicating more mental problems1-Hospitalanxiety and depression scaleUnclearHigh UnclearUnclean UncleanUncleanDepression Intervention11 (4.7 ± 2.6)Control11 (8.3 ± 3.6)AnxietyIntervention11 (7.1 ± 3.7)Control11 (10.4 ± 3.8)Devi 2014 [80]UK6-weekAngina Population66.27 (8.35) in intervention66.20 (10.06) in control 25.5% women 94Activate Your Heart1-Depression 2-Anxiety Higher scorer indicating more mental problems1-Hospitalanxiety and depression scaleLow Low High High Low UncleanDepression Intervention37 (2.00 ± 2.00)Control42 (2.00 ± 4.25)AnxietyIntervention36 (4.14 ± 3.50)Control39 (4.87 ± 3.73)Dodd 2016 [81]Australia 28-week36-week4-monthoverweight or obese29.4 (5.4) for intervention29.6 (5.4) for control Women 2142lifestyle intervention1-Depression2-Anxiety Higher scorer indicating more mental problems1-Edinburgh Postnatal Depression Scale-102-Spielberger’s State-Trait Anxiety InventoryLow Low UnclearUnclearUnclearUnclearDepression 28-weekIntervention976 (6.28 ± 4.53)Control957 (6.12 ± 4.75)36-weekIntervention976 (5.83 ± 4.58)Control957 (5.63 ± 4.72)4-monthIntervention976 (5.34 ± 4.51)Control957 (5.02 ± 4.30)Anxiety28-weekIntervention976 (10.56 ± 3.56)Control957 (10.48 ± 3.66)36-weekIntervention976 (10.64 ± 3.62)Control957 (10.41 ± 3.56)4-month Intervention976 (10.18 ± 3.64)Control957 (10.14 ± 3.50)Forsyth 2015 [82]Australia 3-month patients with depression and anxiety18–84Both (%Women is unknown) 63lifestyle intervention1-Depression 2-Anxiety 3-StressHigher scorer indicating more mental problems1-Depression anxiety and stress scaleHigh High UnclearUnclearHigh Unclear Depression Intervention32 (6.0 ± 6.2)Control31 (5.9 ± 3.5)AnxietyIntervention32 (3.5 ± 3.3)Control31 (3.7 ± 3.5)StressIntervention32 (6.7 ± 5.1)Control31 (8.0 ± 4.8)Furuya 2015 [83]Brazil 6-month Patients following percutaneouscoronary intervention≥1843.3% women 60educational programme1-Depression 2-Anxiety Higher scorer indicating more mental problems1-Hospitalanxiety and depression scaleUnclearLow Low Low UnclearUnclearDepression Intervention30 (5.1 ± 4.4)Control30 (7.6 ± 4.1)Anxiety Intervention30 (5.4 ± 4.8)Control30 (4.7 ± 3.5)Garcia 2023 [84]Spain2-month6-month 12-month treatment-resistant depression≥1869.2% women 65lifestyle modification program 1-Depression Higher scorer indicating more mental problems1-Beck Depression Inventory-IILow Low UnclearUnclearLow Unclear2-monthIntervention34 (17.34 ± 10.8)Control31 (24.87 ± 14.2)6-month Intervention34 (16.85 ± 13.3)Control31 (23.17 ± 17.3)12-monthIntervention34 (19.87 ± 15.9)Control31 (23.33 ± 15.3)Giallo 2014 [85]Australia 2-week6-week Postpartum >18Women 98psychoeducational intervention1-Depression 2-Anxiety 3-StressHigher scorer indicating more mental problems1-Depression anxiety and stress scaleLow UnclearLow UnclearLow Unclear2-week Depression Intervention39 (3.07 ± 4.05)Control59 (5.24 ± 7.01)Anxiety Intervention39 (1.80 ± 3.03)Control59 (2.86 ± 3.96)Stress Intervention39 (10.00 ± 6.18)Control59 (11.87 ± 9.33)6-week Depression Intervention39 (3.85 ± 4.09)Control59 (4.64 ± 5.23)Anxiety Intervention39 (1.95 ± 3.20)Control59 (2.34 ± 3.49)Stress Intervention39 (9.95 ± 7.41)Control59 (10.88 ± 9.12)Glasgow 2006 [86]USA2-month Type 2 diabetes61.5 ± 11.350% women 301lifestyle intervention1-Depression Higher scorer indicating more mental problems1-Patient HealthQuestionnaireUnclearUnclearUnclearUnclearLow UnclearIntervention147 (5.5 ± 5)Control152 (5.5 ± 5.3)Guo 2021 [87]China 3-month 6-month Gestational Diabetes Mellitus≥18Women 320Intensive Lifestyle Modification1-Stress Higher scorer indicating more mental problems1-perceived stress scaleLow Low High LowLow Unclear3-month Intervention160 (24.22 ± 7.93)Control160 (24.53 ± 6.72)6-monthIntervention160 (24.18 ± 7.33)Control160 (24.60 ± 5.47)Han 2020 [88]Hong Kong15-weekmajor depressive disorder47.06 (9.54) in intervention45.44 (8.25) in control Both (%Women is unknown) 33Dejian mind-body intervention1-Depression Higher scorer indicating more mental problems1-Hamilton Psychiatric Rating Scalefor Depression (HRSD)2-Beck Depression InventoryLow Low Low Low UnclearUnclearHRSDIntervention17 (6.50 ± 4.31)Control16 (9.75 ± 4.16))BDIIntervention17 (17.94 ± 12.70)Control16 (24.79 ± 14.91))Heutink 2012 [89]The NetherlandsPost-intervention3-month spinal cord injury≥1830.06% women 61Multidisciplinarycognitive behavioral program1-Anxiety Higher scorer indicating more mental problems1-Hospitalanxiety and depression scaleUnclear Unclear Unclear Unclear Low Unclear Post-interventionIntervention31 (5.6 ± 3.6)Control30 (5.7 ± 3.4)3-month Intervention31 (5.9 ± 3.6)Control30 (5.6 ± 3.6)Hilmarsdóttir 2021 [90]Iceland6-month type 2 diabetes mellitus25–7063.3% women 30Sidekick Health smartphone app1-Depression 2-Anxiety Higher scorer indicating more mental problems1-Hospitalanxiety and depression scaleUnclear Low High Low UnclearUnclearDepression Intervention15 (3.3 ± 3.0)Control15 (4.2 ± 4.6)Anxiety Intervention15 (4.1 ± 3.8)Control15 (5.5 ± 4.7)Holt 2019 [91]UK3-month 12-month Schizophrenia ≥1849% women 412structured education lifestyle program1-Depression Higher scorer indicating more mental problems1-Patient HealthQuestionnaireLow High High Low UnclearUnclear3-month Intervention178 (10.3 ± 6.3)Control180 (10.1 ± 7.1)12-monthIntervention167 (9.9 ± 7.0)Control173 (9.6 ± 6.6)Hwang 2019 [92]Korea 4-weeknurses employedUnspecified 94.6% women 56Stress-Management Program1-Depression 2-Anxiety 3-Stress Higher scorer indicating more mental problems1-Patient HealthQuestionnaire2-Generalized Anxiety Disorder-73-Perceived Stress Scale-10UnclearUnclearUnclearUnclearUnclearUnclearDepression * Intervention26 (6.46 ± 4.99)Control30 (6.93 ± 4.98)Anxiety Intervention26 (4.23 ± 4.38)Control30 (5.40 ± 4.38)StressIntervention26 (18.50 ± 3.56)Control30 (19.16 ± 3.56)Ihle-Hansen 2014 [93]Norway 12-month Stroke 72.6 (11.2 in intervention70.6 (13.6) in control 46.7% women 195multifactorial risk factor intervention program1-Depression 2-Anxiety Higher scorer indicating more mental problems1-Hospitalanxiety and depression scaleUnclearUnclearLow Low Low UnclearDepression Intervention98 (2.91 ± 2.63)Control97 (3.49 ± 3.02)Anxiety Intervention97 (3.10 ± 2.83)Control97 (3.95 ± 3.50)Imayama 2011 [94]USA12-month overweight/obesepostmenopausal women50–75Women 204diet and/or exercise interventions1-Depression 2-Anxiety 3-StressHigher scorer indicating more mental problems1-Brief Symptom Inventory-182-Perceived Stress ScaleLow UnclearUnclearLow Low UnclearDepression Intervention117 (46.2 ± 8.2)Control87 (48.4 ± 9.6)Anxiety Intervention117 (43.5 ± 6.4)Control87 (45.3 ± 8.7)StressIntervention117 (2.66 ± 2.27)Control87 (3.89 ± 2.75)Inouye 2014 [95]USA6-month at risk for diabetes≥30Both %Women is unknown 40Lifestyle Intervention1-Depression Higher scorer indicating more mental problems1- Center for Epidemiologic Studies Depression ScaleUnclearUnclearUnclearUnclearUnclearUnclearIntervention22 (12.21 ± 10.97) Control18 (13.52 ± 11.28)Ip 2021 [96]China 6-week12-weekmoderate to severe depression≥1883.9% women 31group-based lifestyle medicine1-Depression 2-Anxiety 3-StressHigher scorer indicating more mental problems1-Patient HealthQuestionnaire2-Depression anxiety and stress scaleLow Low High Low Low Unclear6-weekPHQ-9-depressionIntervention16 (7.4 ± 2.3)Control15 (9.5 ± 3.7)DASS—DepressionIntervention16 (7.0 ± 3.7)Control15 (12.9 ± 7.8)AnxietyIntervention16 (4.3 ± 2.5)Control15 (11.1 ± 8.0)Stress Intervention16 (13.5 ± 7.7)Control15 (17.1 ± 9.6)12-weekPHQ-9-depression Intervention16 (7.5 ± 3.6)Control15 (10.2 ± 3.8)DASS—DepressionIntervention16 (9.5 ± 7.8)Control15 (13.3 ± 9.0)Anxiety Intervention16 (6.5 ± 2.8)Control15 (10.7 ± 7.3)Stress Intervention16 (11.5 ± 6.9)Control15 (16.9 ± 9.6)Jonsdottir 2015 [99]Iceland6-month obstructive pulmonarydisease45–6554% women 100self-management programme1-Depression 2-Anxiety Higher scorer indicating more mental problems1-Hospitalanxiety and depression scaleLow UnclearLow UnclearLow UnclearDepression Intervention46 (3.28 ± 3.30)Control49 (3.92 ± 3.28)Anxiety Intervention48 (6.60 ± 4.26)Control52 (7.25 ± 3.61)Kelly 2020 [100]Australia 12-week16-weekconsumersof a community mental health service18–6558% women 43peer delivered healthy lifestyle intervention1-Depression Higher scorer indicating more mental problems1-Patient HealthQuestionnaireLow Low UnclearLow UnclearUnclear12-weekIntervention13 (11.62 ± 6.55)Control 14 (12.79 ± 7.54)16-weekIntervention16 (10.50 ± 6.13)Control16 (11.94 ± 6.12)Kieffer 2013 [101]USAUnknown Pregnant ≥18Women 275Healthy Lifestyle Intervention1-Depression Higher scorer indicating more mental problems1-Center for Epidemiologic Studies Depression ScaleLow Low UnclearLow Low UnclearIntervention138 (11.24 ± 7.98)Control137 (12.71 ± 7.84)Kim 2011 [102]Korea12-weekBreast Cancer26–69Women 45Matched Exercise and Diet1-Depression 2-Anxiety Higher scorer indicating more mental problems1-Hospitalanxiety and depression scaleUnclearUnclearUnclearUnclearLow UnclearDepressionIntervention23 (3.32 ± 2.58)Control22 (5.85 ± 3.65)Anxiety Intervention23 (3.97 ± 2.30)Control22 (5.46 ± 2.76)Koch 2021 [103]Germany12-week24-week48-weekUlcerativeColitis18–74Unknown 97Lifestyle Modification1-Depression2-Anxiety 2-Stress Higher scorer indicating more mental problems1-Hospitalanxiety and depression scale1-perceived stress scaleLow UnclearUnclearUnclearLow UnclearDepression 12-weekIntervention47 (4.74 ± 3.39)Control50 (5.81 ± 3.91)24-weekIntervention47 (5.57 ± 3.52)Control50 (6.18 ± 3.59)48-weekIntervention47 (4.45 ± 3.46)Control50 (4.74 ± 3.36)Anxiety12-weekIntervention47 (6.67 ± 3.84)Control50 (7.45 ± 3.55)24-weekIntervention47 (7.61 ± 4.26)Control50 (7.55 ± 3.48)48-weekIntervention47 (6.46 ± 3.98)Control50 (6.55 ± 3.20)Stress12-weekIntervention47 (14.00 ± 6.38)Control50 (18.59 ± 6.89)24-weekIntervention47 (15.76 ± 6.44)Control50 (18.47 ± 6.29)48-weekIntervention47 (13.75 ± 7.20)Control50 (16.05 ± 6.80)Kwon 2015 [105]Korea4-weekCommunity Dwelling≥6559.5% women 93Wheel of Wellness counseling intervention1-DepressionHigher scorer indicating more mental problems1-Patient HealthQuestionnaireLow Low High Unclear UnclearUnclearIntervention43 (4.51 ± 4.59)Control46 (5.02 ± 5.47)Lee 2015 [106]Korea 6-month obstructive pulmonarydisease40–808.6% women 151nurse-led problem-solving therapy1-Depression Higher scorer indicating more mental problems1- Center for Epidemiologic Studies Depression ScaleUnclear UnclearUnclear Unclear High UnclearIntervention78 (15.9 ± 8.0)Control73 (17.2 ± 8.0)Leemrijse 2016 [107]the Netherlands6-month patientswith recent coronary event18–8019% women 374Hartcoach1-Depression 2-Anxiety 
Higher scorer indicating more mental problems1-Hospitalanxiety and depression scaleLow UnclearLowUnclearLow UnclearDepression Intervention145 (3.31 ± 3.71)Control167 (3.83 ± 3.64)AnxietyIntervention145 (3.95 ± 3.59)Control167 (4.88 ± 4.00)Lund 2012 [108]Norway9-month stroke survivors75 (7.2 in intervention79 (6.5) in control51.2% women 204lifestyle course1-Depression 2-Anxiety Higher scorer indicating more mental problems1-Hospitalanxiety and depression scaleLow Low UnclearLow UnclearUnclearDepression Intervention39 (3.4 ± 2.7)Control47 (4.2 ± 3.4)AnxietyIntervention39 (3.1 ± 3.4)Control47 (4.4 ± 4.0)María Nápoles 2020 [109]USA3-month6-monthbreast cancer survivors28–88Women 153stressmanagement intervention1-Depression 2-Anxiety 3-Stress Higher scorer indicating more mental problems1-Patient HealthQuestionnaire2-Brief Symptom Inventory scales3-Perceived Stress ScaleLow Low UnclearLow Low UnclearDepression 3-month Intervention76 (6.81 ± 5.31)Control77 (6.97 ± 5.12)6-month Intervention76 (6.96 ± 5.62)Control77 (6.44 ± 5.15)Anxiety 3-monthIntervention76 (0.63 ± 0.61) Control77 (0.65 ± 0.70)6-monthIntervention76 (0.52 ± 0.53)Control77 (0.70 ± 0.64)Stress3-month Intervention76 (14.45 ± 6.63)Control77 (14.08 ± 7.35)6-monthIntervention76 (14.70 ± 6.14)Control77 (15.14 ± 6.28)Martín 2014 [110]Spain 6-month Fibromyalgia50.12 ± 9.0793.5% women 110Interdisciplinary PSYMEPHYTreatment1-Anxiety Higher scorer indicating more mental problems1-Fibromyalgia Impact QuestionnaireLow UnclearUnclearUnclearUnclearUnclearIntervention54 (13.41 ± 4.31)Control56 (12.75 ± 4.55)Mayer-Davis 2018 [111]USA18-monthType 1 diabetes13–1638.8% women 99Flexible Lifestyles for Youth1-Depression Higher scorer indicating more mental problems1- Center for Epidemiologic Studies Depression ScaleLow Low UnclearHigh Low UnclearIntervention118 (6∙63 ± 7∙12)Control123 (8∙46 ± 7∙08)Mensorio 2019 [112]Spain3-monthObesity and hypertension18–65Unknown 106Living Better1-Depression 2-Anxiety 3-StressHigher scorer indicating more mental problems1-Depression anxiety and stress scaleLow Unclean UnclearUnclearLow UnclearDepression Intervention43 (2.88 ± 3.6)Control48 (2.79 ± 3.5)AnxietyIntervention43 (1.73 ± 2.6)Control48 (3.04 ± 3.4)StressIntervention43 (3.40 ± 2.9)Control48 (5.20 ± 4.1)Michalsen 2005 [113]Germany 12-month CoronaryArtery Disease59.4 ± 8 8.622.8% Women 101lifestyle modification program1-Depression 2-Anxiety 3-StressHigher scorer indicating more mental problems1-BeckDepression Inventory2-SpielbergerState-Trait Anger Expression Inventory3-Cohen Perceived Stress ScaleLow Low UnclearUnclearLow UnclearDepression Intervention48 (6.4 ± 4.2)Control53 (7.6 ± 4.7)State anxietyIntervention48 (36.5 ± 8.8)Control53 (36.2 ± 7.6)Trait anxietyIntervention48 (35.7 ± 8.3)Control53 (37.5 ± 10.0)StressIntervention48 (19.1 ± 7.6)Control53 (21.7 ± 7.7)Moncrieft 2016 [144]USA6-month 12-monthType 2 Diabetes18–7071.2% women 111Lifestyle Modification1-Depression Higher scorer indicating more mental problems1-Beck Depression Inventory-IILow Low Unclean Low Low Unclear6-month Intervention42 (10.75 ± 7.76)Control48 (16.09 ± 9.15)12-monthIntervention41 (9.85 ± 8.86)Control46 (16.00 ± 10.80)Moore 2011 [158]Australia6-month At risk of type 2 diabetes61.3 ± 11.159% women 307group-basedlifestyle intervention1-Depression 2-Anxiety Higher scorer indicating more mental problems1-Depression anxiety and stress scaleHigh UncleanUncleanUncleanUncleanUncleanDepression Intervention167 (5.32 ± 7.03)Control85 (4.87 ± 6.78)Anxiety Intervention167 (4.57 ± 5.84)Control85 (3.56 ± 4.31)Morales-Fernández 2021 [159]Spain 3-month6-month 9-monthnon-malignant pain45–61 percentile67.7% women 279nurse-led intervention1-Depression2-Anxiety Higher scorer indicating more mental problems1-Patient HealthQuestionnaire2-Generalized Anxiety Disorder ScaleLow Low High Low Low UnclearDepression 3-monthIntervention174 (10.06 ± 5)Control105 (11.31 ± 6.19)6-month Intervention174 (10.16 ± 5.11)Control105 (11.61 ± 5.81)9-monthIntervention174 (10.43 ± 5.29)Control105 (12.66 ± 5.99)Anxiety3-monthIntervention174 (8.43 ± 4.76)Control105 (9.1 ± 5.31)6-month Intervention174 (7.72 ± 4.73)Control105 (8.95 ± 4.92)9-monthIntervention174 (7.51 ± 4.77)Control105 (9.16 ± 4.7)Moseley 2009 [115]Australia Post-Program6-week Adolescent15.6 ± 0.666.7% women 81School-Based Intervention1-Depression Higher scorer indicating more mental problems1-Depression anxiety and stress scaleUnclearUnclearUnclearUnclearUnclearUnclearPost-ProgramIntervention21 (10.0 ± 7.7)Control12 (14.2 ± 11.8))6-weekIntervention17 (12.9 ± 7.3)Control13 (13.9 ± 10.7)Mountain 2017 [116]UK6-month 24-montholderadults≥6568.05% women 262occupation-based lifestyle intervention1-DepressionHigher scorer indicating more mental problems1-Patient HealthQuestionnaireLow Low High Low Low Unclear6-month Intervention133 (3.8 ± 4.2)Control122 (3.4 ± 4.3)24-monthIntervention122 (3.8 ± 4.6)Control114 (4.0 ± 4.8)Murawski 2019 [117]Australia3-month 6-monthAdults with insufficient physical activity/poor sleep quality18−5580% women 160physical activity and sleep quality1-Depression 2-Anxiety 3-StressHigher scorer indicating more mental problems1-Depression anxiety and stress scaleLow Low UnclearUnclearLow UnclearDepression 3-month Intervention59 (10.6 ± 7.62)Control65 (12.6 ± 7.97)6-monthIntervention34 (10.9 ± 8.01)Control53 (13.3 ± 9.49)Anxiety3-month Intervention59 (6.4 ± 3.65)Control65 (7.5 ± 5.04)6-monthIntervention34 (5.9 ± 3.54)Control53 (8.9 ± 4.70)Stress3-month Intervention59 (13.6 ± 4.20)Control65 (15.4 ± 4.97)6-monthIntervention34 (13.0 ± 5.75)Control53 (16.3 ± 5.24)Nie 2019 [118]China3-month 6-month9-month 12-monthcoronary artery disease18–8027.5% women 284lifestyle improving program1-Depression 2-Anxiety Higher scorer indicating more mental problems1-Hospitalanxiety and depression scaleLow Low High Low Low UnclearDepression3-monthIntervention142 (9.85 ± 2.48)Control142 (9.71 ± 2.94)6-monthIntervention142 (9.44 ± 2.84)Control142 (9.48 ± 3.06)9-month Intervention142 (8.57 ± 2.81)Control142 (9.13 ± 3.22)12-monthIntervention142 (8.21 ± 3.03)Control142 (9.08 ± 3.30)Anxiety3-monthIntervention142 (9.37 ± 2.74)Control142 (9.63 ± 2.38)6-monthIntervention142 (8.82 ± 2.51)Control142 (9.21 ± 2.52)9-month Intervention142 (8.26 ± 2.24)Control142 (8.93 ± 2.36)12-monthIntervention142 (7.80 ± 2.38)Control142 (8.88 ± 2.37)O’Neill 2015 [119]UK6-monthprostate cancer69.7 ± 6.8 in intervention69.9 ± 7.0 in control Men 94diet and physical activity1-StressHigher scorer indicating more mental problems1-Perceived Stress ScaleLow Low Unclean High Low UnclearIntervention47 (10.5 ± 6.9)Control47 (11.2 ± 10.2)O’Reilly 2016 [120]Australia 12-month Gestational Diabetes≥18Women 573group-based lifestyle modification1-Depression
Higher scorer indicating more mental problems1-Patient HealthQuestionnaireLowUnclear Unclear Unclear LowUnclearIntervention284 (4.41 ± 4.38)Control289 (4.39 ± 4.25)Phelan 2014 [121]USA6-month 12-monthpregnancy>18Women 401behavioral intervention1-Depression2-Stress Higher scorer indicating more mental problems1-Edinburgh Postnatal Depression Scale2-Perceived Stress ScaleLow Unclear LowLow Low UnclearDepression 6-monthIntervention128 (5.1 ± 4.2)Control133 (4.4 ± 3.6)12-monthIntervention128 (5.6 ± 4.2)Control133 (4.9 ± 4.1)Stress6-monthIntervention128 (8.3 ± 3.0)Control133 (7.8 ± 2.9)12-monthIntervention128 (8.4 ± 2.8)Control133 (8.1 ± 2.9)Psarraki 2021 [122]GreeceUnknown major depressive disorder18–6583.9% women69Pythagorean Self-Awareness1-Depression 2-Anxiety 3-StressHigher scorer indicating more mental problems1-Depression anxiety and stress scale2-Beck Depression InventoryLow UnclearHigh High Unclear Unclear Depression Intervention30 (13.31 ± 9.71)Control32 (18.41 ± 12.66)AnxietyIntervention30 (14.77 ± 11.07)Control32 (15.19 ± 12.59)StressIntervention30 (14.89 ± 9.69)Control32 (19.40 ± 10.08)BDIIntervention30 (14.70 ± 9.77)Control32 (22.28 ± 13.45)Sacco 2009 [123]USA6-monthtype 2 diabetes18–65Both %Women is unknown 62regular telephone intervention1-DepressionHigher scorer indicating more mental problems1-Patient HealthQuestionnaireHigh Unclear Unclear Unclean Low Unclear Intervention31 (14.74 ± 5.96)Control 31 (16.87 ± 7.39)Sanaati 2017 [124]Iran 8-weekPregnancy 27.5 (4.9) in intervention27.7 (4.9) in control Women125lifestyle-based education1-Depression2-Anxiety Higher scorer indicating more mental problems1-Edinburgh Postnatal Depression Scale2-Spielberger State-Trait Anxiety InventoryLow Low High Low Low Unclear Depression Intervention62 (4.6 ± 3.5)Control63 (7.5 ± 3.7)State anxietyIntervention62 (34.4 ± 6.4)Control63 (39.1 ± 9.2)Trait anxietyIntervention62 (33.4 ± 7.1)Control63 (39.0 ± 8.3)Saxton 2014 [125]UK6-month breast cancer55.8 (10.0) in intervention55.3 (8.8) in control Women 85pragmatic lifestyle intervention1-Depression 2-Stress Higher scorer indicating more mental problems1-Beck Depression Inventory2-Perceived Stress ScaleLow Low UnclearLow LowUnclear DepressionIntervention44 (5.1 ± 4.9)Control41 (7.9 ± 6.0)StressIntervention44 (18.2 ± 7.7)Control41 (19.5 ± 6.8)Sebregts 2005 [126]the NetherlandsPost-intervention 9-month acute myocardial infarction or coronary artery bypass grafting55.6 [8.0 in intervention55.2 [9.7] in control Both %Women is unknown 184hort behavior modification program1-Depression Higher scorer indicating more mental problems1-Beck Depression InventoryUnclean Low LowUnclearLowUnclear Post-intervention Intervention83 (7.7 ± 6.0)Control75 (5.8 ± 4.9)9-monthIntervention83 (6.9 ± 4.8)Control75 (5.8 ± 5.1)Serrano Ripoll 2015 [127]Spain 6-month12-month PrimaryCare patientsIQR 40–6182%women 273Lifestyle change recommendations1-Depression 2-Anxiety Higher scorer indicating more mental problems1-Beck Depression Inventory2-Spielberger State-Trait Anxiety InventoryUnclearLow Low Low LowUnclear Depression6-monthIntervention 106 (18.2 ± 9.98)Control120 (16.6 ± 12.57)12-monthIntervention 95 (17.3 ± 9.44)Control99 (16.1 ± 11.42)Anxiety6-monthIntervention 106 (70.9 ± 25.47)Control120 (62.3 ± 27.10)12-monthIntervention 95 (67.8 ± 20.88)Control99 (61.0 ± 29.95)Sheean 2021 [128]USA12-weekmetastatic breast cancer≥18Women 35lifestyle intervention1-Depression 2-Anxiety 3-Stress Higher scorer indicating more mental problems1-Hospitalanxiety and depression scale2-Perceived Stress ScaleLow Low Unclear Low UnclearUnclearDepression Intervention17 (3.2 ± 3.1)Control18 (3.4 ± 3.6)AnxietyIntervention17 (5.8 ± 3.9)Control18 (4.6 ± 5.1)StressIntervention17 (14.0 ± 6.2)Control18 (12.3 ± 9.4)Sorensen 1999 [129]NorwayUnknown elevatedrisk factors for cardiovascular disease41–50Both %Women is unknown 219exercise and diet1-Depression2-Anxiety Higher scorer indicating more mental problems1-General Health Questionnaire2- SymptomCheck List-90 (SCL-90)Unclear Unclear Unclear Unclear Unclear Unclear Depression Intervention67 (2.0 ± 1.7)Control43 (2.7 ± 2.9)AnxietyIntervention67 (4.8 ± 5.2)Control43 (7.5 ± 6.8)Speyer 2016 [130]DenmarkUnknown Schizophreni/abdominal obesity38.6 ± 12.456.1 women 428lifestyle coaching1-Stress Higher scorer indicating more mental problems2-Perceived Stress ScaleLow Low Low Low Low UnclearIntervention138 (26.8 ± 7.8)Control148 (25.5 ± 7.4)Sylvia 2019 [131]USA20-weekbipolar disorder18–6565.8% women 38Nutrition exercisewellness treatment1-DepressionHigher scorer indicating more mental problems1-Montgomery Asberg DepressionRating Scale2-Clinical Global Impression ScaleUnclear Unclear Unclear Low Low Unclear CGIIntervention19 (2.3 ± 0.9)Control19 (2.4 ± 1.0)MADRSIntervention19 (8.1 ± 6.7)Control19 (9.6 ± 8.2)Takeda 2020 [132] Japan7-month Elderly 77.03 (8.08 in intervention75.51 (6.55) in control 88.2% women 127lifestyle development program1-Depression Higher scorer indicating more mental problems1–15-item GeriatricDepression ScaleUnclear Unclear Unclear Unclear Unclear Unclear Intervention60 (4.20 ± 2.93)Control67 (4.64 ± 3.61)Toobert 2007 [133]USA6-month12-month24-month type 2diabetes61.1 (8.0) in intervention60.7 (7.8) in control Women 279Mediterranean lifestyle program1-Depression2-StressHigher scorer indicating more mental problems1-Center for Epidemiologic Studies Depression Scale2-Perceived Stress ScaleUnclear Unclear Unclear Unclear Low UnclearDepression 6-monthIntervention163 (13 ± 11)Control116 (15 ± 12)12-monthIntervention163 (15 ± 11)Control116 (14 ± 9)24-monthIntervention163 (12 ± 11)Control116 (14 ± 10)Stress6-monthIntervention163 (2.5 ± 0.62)Control116 (2.6 ± 0.59)12-monthIntervention163 (2.6 ± 0.66)Control116 (2.6 ± 0.58)24-monthIntervention163 (2.4 ± 0.64)Control116 (2.6 ± 0.61)Tousman 2011 [134]USA2-month Asthma51.4 (14.7) in intervention55.0 (10.0)68.9% women 45behavior modification procedure1-Depression Higher scorer indicating more mental problems1-Geriatric Depression ScaleUnclear Unclear Unclear Unclear Unclear Unclear Intervention21 (1.8 ± 2.1)Control24 (1.9 ± 2.1)Trento 2020 [135]Italy 4-year type 2diabetes62.6 ± 7.5 in intervention ± 9.1 in control 36% women 50Self-management education1-Depression 2-Anxiety Higher scorer indicating more mental problems1-Hospitalanxiety and depression scaleUnclean Low High High Low UnclearDepression Intervention24 (4.5 ± 3.86)Control25 (3.44 ± 2.95)AnxietyIntervention24 (4.83 ± 3.25)Control25 (5.28 ± 3.45)Tsai 2021 [136]Taiwan2-weekat-risk mental state20–3555.4% women 92Health-Awareness-Strengthening Lifestyle1-Anxiety Higher scorer indicating more mental problems1-State and Trait Anxiety InventoryLow UnclearUnclearUnclearLow UnclearState AnxietyIntervention46 (42.5 ± 7.7)Control46 (47.7 ± 9.5)Trait AnxietyIntervention46 (52.0 ± 7.0)Control46 (56.0 ± 7.0)Ural 2021 [137]Turkey6-week gestational diabetes mellitus≥18Women 88health-promoting lifestyle education1-Depression Higher scorer indicating more mental problems1-Center for Epidemiologic Studies Depression ScaleHigh High UnclearUnclearUnclearUnclearIntervention46 (14.93 ± 9.39)Control42 (15.74 ± 8.54)Van Dammen 2019 [138]the Netherlands5-yearObesity andinfertility18–39Women 577lifestyle intervention1-Depression 2-Anxiety 3-StressHigher scorer indicating more mental problems1-Hospitalanxiety and depression scale 2-Perceived Stress Scale-10UnclearUnclearUnclearUnclearUnclearUnclearDepressionIntervention84 (7.7 ± 3.66)Control94 (7.7 ± 2.90)AnxietyIntervention84 (8.0 ± 3.66)Control94 (8.2 ± 2.90)StressIntervention52 (14.4 ± 6.48)Control63 (13.7 ± 4.76)van der Wulp 2012 [139]the Netherlands3-month 6-monthtype 2diabetesUnknown45.4% women 119peer-led self-management1-Depression Higher scorer indicating more mental problems1-Center for Epidemiologic Studies Depression ScaleUnclearUnclearUnclearUnclearLow Unclear3-month Intervention59 (10.60 ± 8.03)Control60 (11.20 ± 8.52)6-monthIntervention59 (8.64 ± 8.56)Control60 (12.07 ± 9.55)Wang 2014 [140]USA4-month 12-monthtype 2 diabetes≥1876.6 women 252Diabetes Self-Management1-Depression Higher scorer indicating more mental problems1-Center for Epidemiologic Studies Depression ScaleLow Low UnclearLow UnclearUnclear4-month Intervention117 (17.5 ± 13.0)Control112 (21.8 ± 12.4)12-monthIntervention109 (18.5 ± 13.0)Control107 (22.6 ± 13.4)Wang 2017 [141]China1-month 3-month Metabolic syndrome24–7850.9% women 173lifestyle intervention program1-Depression Higher scorer indicating more mental problems1-Hospitalanxiety and depression scaleLow Low UnclearLow Low Unclear1-month Intervention86 (3.23 ± 2.71)Control87 (3.94 ± 3.49)3-monthIntervention86 (2.13 ± 2.06)Control87 (3.43 ± 2.96)Williams 2018 [142]Australia26-week Low Back Pain56.7 ± 13.459.1% women 159Healthy Lifestyle Intervention1-Depression 2-Anxiety 3-StressHigher scorer indicating more mental problems1-Depression anxiety and stress scaleLow Low Unclear Low Low Low Depression Intervention43 (13.1 ± 11.2)Control61 (11.9 ± 11.1)Anxiety Intervention43 (9.8 ± 8.3)Control61 (9.4 ± 9.0)StressIntervention43 (14.3 ± 10.7)Control61 (13.8 ± 11.1)Wong 2021 [143]China 9-weekmoderate level of depressive symptoms32.9 ± 12.584.8% women 79Lifestyle Medicine1-Depression2-Anxiety Higher scorer indicating more mental problems1-Patient HealthQuestionnaire2-General Anxiety DisorderLow UnclearHigh Unclean UnclearUnclearDepression Intervention39 (8.8 ± 3.8)Control40 (11.6 ± 4.7)AnxietyIntervention39 (7.8 ± 3.2)Control40 (11.5 ± 4.6)Sample size and *p* valueAdvocat 2016 [65]Australia6-month Parkinson’s disease18–7557.9% women 48Mindfulness-based lifestyle1-Depression 2-Anxiety 3-StressHigher scorer indicating more mental problems1-Depression anxiety and stress scaleLow Low Low Low High Unclear Depression Intervention23Control25*p* = 0.54AnxietyIntervention23Control25*p* = 0.38StressIntervention23Control25*p* = 0.04Brown 2006 [71]UK6-weekSerious mental illness18–6585.7% women 17health promotion sessions1-Depression 2-Anxiety Higher scorer indicating more mental problems1-Hospitalanxiety and depression scaleUnclearLow UnclearLow Low Unclear Depression Intervention7Control10*p* = 0.080Anxiety Intervention7Control10*p* = 0.190Gallagher 2014 [144]Australia16-weekoverweight withheart disease and diabetes63.2 ± 8.6940% women 147Group-based lifestyle intervention1-Depression 
Higher scorer indicating more mental problems1-Hospitalanxiety and depression scaleLowUnclearUnclearUnclearLowUnclear Intervention75Control58*p* = 0.014Gaudel 2021 [145]Nepal1-month coronary artery disease>1824.1% women 224lifestyle-related risk factor modification intervention1-Stress Higher scorer indicating more mental problems1-Perceived Stress Scale-10Low Unclear High UnclearUnclearUnclear Intervention98Control98*p* = 0.000Goracci 2016 [146]Italy 12-month Recurrent depression>1880% women 160lifestyle intervention1-Depression Higher scorer indicating more mental problems1-Patient HealthQuestionnaireUnclear Unclear Unclear Unclear Low Unclear Intervention81 Control79 *p* = 0.29Islam 2013 [97]USA6-month atrisk for diabetes18–7564.3% women 35Healthy Lifestyles1-Depression 2-Anxiety Higher scorer indicating more mental problems1-Patient HealthQuestionnaire2-Generalized Anxiety Disorder ScaleUnclear Unclear Unclear Unclear Unclear Unclear DepressionIntervention21Control14*p* = 0.43Anxiety Intervention21Control14*p* = 0.15Jiskoot 2020 [147]The Netherlands12-month Polycystic Ovary Syndrome18–38Women 120lifestyle intervention1-Depression Higher scorer indicating more mental problems1-Beck Depression Inventory-IILow UnclearUnclearUnclearLow UnclearIntervention60Control60*p* = 0.045Jørstad 2016 [99]theNetherlands12-month Acutecoronary syndrome18–8022.5% women 120nurse-coordinated preventionprogramme1-Depression Higher scorer indicating more mental problems1-Beck Depression Inventory-IILow UnclearUnclearUnclearUnclearUnclear Intervention54Control66*p* = 0.03Kokka 2019 [104]Greece8-week Intimate PartnerViolence18–70Women60stressmanagement program1-Depression 2-Anxiety 3-StressHigher scorer indicating more mental problems1-Depression anxiety and stress scaleLow Unclean High High Low LowDepression Intervention30Control30*p* = 0.000Anxiety Intervention30Control30*p* = 0.000StressIntervention30Control30*p* = 0.000Lorig 2009 [148]USA6-month type 2 diabetes24–93Both %Women is unknown294Peer-Led Diabetes Self-management1-Depression Higher scorer indicating more mental problems1-Patient HealthQuestionnaireLow Unclean High Unclean Low Unclear Intervention161Control133*p* = 0.000Lovell 2014 [149]UK6-month 12-month psychosis16–3540% women 105Healthy LivingIntervention1-Depression Higher scorer indicating more mental problems1-Calgary Depression Scale.Low UncleanHigh Low Low Unclear 6-month Intervention46Control39*p* = 0.9812-monthIntervention48Control42*p* = 0.65Mitchell 2014 [160]UK6-month chronic obstructive pulmonarydisease69 ± 8.0 in intervention69 ± 10.1 in control45.1% women 184self-management programme1-Depression 2-Anxiety Higher scorer indicating more mental problems1-Hospitalanxiety and depression scaleLow Low High Low Low Unclear DepressionIntervention89Control95*p* = 0.27AnxietyIntervention89Control95*p* = 0.04Pelekasis 2016 [150]Greece8-weekBreastCancer 18–75Women61stress management1-Depression 2-Anxiety 3-StressHigher scorer indicating more mental problems1-Depression anxiety and stress scaleLow Unclear Unclear Unclear Low Unclear Depression Intervention25Control28*p* = 0.01AnxietyIntervention25Control28*p* = 0.005StressIntervention25Control28*p* = 0.002Przybylko 2021 [151]Australia and New Zealand12-week24-weekGeneral population 46.97 ± 14.569.9% women 320Onlineinterdisciplinary intervention1-Depression 2-Anxiety 3-StressHigher scorer indicating more mental problems1-Depression anxiety and stress scaleLow Unclean High High Low Unclear Depression 12-weekIntervention159 Control162*p* = 0.00224-weekIntervention159 Control162*p* = 0.005Anxiety 12-weekIntervention159 Control162*p* = 0.00024-weekIntervention159 Control162*p* = 0.035Stress12-weekIntervention159 Control162*p* = 0.00124-weekIntervention159 Control162*p* = 0.000Rosal 2005 [152]USA3-month6-month type 2 diabetes45–8280% women 25Diabetes Self-Management1-Depression Higher scorer indicating more mental problems1- Center for Epidemiologic Studies Depression ScaleUnclear Unclear Unclear Unclear Unclear Unclear 3-monthIntervention15Control10*p* < 0.056-monthIntervention15Control10*p* = 0.01Ruusunen 2012 [153]FinlandUnknown overweight or obese/glucose tolerance40–6457.9% women 140lifestyle intervention1-Depression Higher scorer indicating more mental problems1-Beck Depression InventoryUnclear Unclear Unclear Unclear Unclear Unclear Intervention69Control71*p* = 0.965Samuel-hodge 2017 [154]USA20-weekoverweight orobesity and type 2 diabetes21–7581% women 108Lifestyle Support1-DepressionHigher scorer indicating more mental problems1-Patient HealthQuestionnaireUnclear Unclear High Low UnclearUnclear Intervention34Control16*p* = 0.01Skrinar 2005 [155]USA12-week Serious PsychiatricDisabilities18–55Unknown 20healthy lifestyle1-Depression 2-Anxiety Higher scorer indicating more mental problemsSymptomCheck List-90 (SCL-90)Unclear Unclear Unclear Unclear Unclear Unclear DepressionIntervention9 Control11*p* = 0.09AnxietyIntervention9 Control11*p* = 0.59Surkan 2012 [156]USAUnknown Postpartum18–44Women 403Health Promotion Intervention1-Depression Higher scorer indicating more mental problems1- Center for Epidemiologic Studies Depression ScaleUnclear Unclear Unclear Unclear LowUnclear Intervention203 Control200*p* = 0.046Ye 2016 [157]China2-month6-month12-monthBreast cancer UnknownWomen 204mentor-based program1-Depression2-Anxiety Higher scorer indicating more mental problems1-Hospitalanxiety and depression scaleUnclear Unclear Unclear Unclear Unclear Unclear Depression 2-month Intervention99Control101*p* = 0.00196-monthIntervention95Control92*p* = 0.00012-monthIntervention90Control81*p* = 0.000Anxiety2-monthIntervention100Control102*p* = 0.04856-monthIntervention96Control91*p* = 0.00012-monthIntervention91Control79*p* = 0.000* calculated by author(s).


### 3.2. Quality Assessment of Studies

A qualitative evaluation of the eligible studies was conducted, based on the results of the qualitative evaluation listed in Table 1.

### 3.3. Lifestyle Intervention and Depression

A meta-analysis of 89 randomized clinical trials of lifestyle interventions on depression indicated that lifestyle interventions lead to a reduction in depression, according to which Hedges’s g was equal to −0.21 with 95% confidence interval −0.26, −0.15 (Z = −7.12; *p* < 0.001; *I*^2^ = 56.57) (not shown in the figure).

### 3.4. Sub-Group Analysis for Lifestyle Intervention and Depression

Table 2 shows the results of the meta-analysis of lifestyle interventions for depression across different populations. In the cancer population, lifestyle interventions led to a reduction in depression (Hedges’s g = −0.34; 95% CI −0.59, −0.08 [Z = −2.54; *p* = 0.011; *I*^2^ = 56.23%]). In the depressed population, lifestyle interventions led to a reduction in depression [Hedges’ g = −0.44; 95% CI −0.62, −0.26; Z = −4.82; *p* < 0.001; *I*^2^ = 40.46%). In the diabetes/at-risk diabetes population, lifestyle interventions led to a reduction in depression [Hedges’ g = −0.15; 95% CI −0.27, −0.03 (Z = −2.43; *p* = 0.015; *I*^2^ = 56.51%). In the heart-related disease population, lifestyle interventions led to a reduction in depression (Hedges’s g = −0.19; 95% CI −0.34, −0.04 [Z = −2.44; *p* = 0.015; *I*^2^ = 39.52%]). In the metabolic syndrome population, lifestyle interventions led to a reduction in depression [Hedges’ g = −0.74; 95% CI −1.27, −0.21 (Z = −2.44; *p* = 0.006; *I*^2^ = 69.66%). In obstructive pulmonary disease, older adults, and the overweight/obese population, lifestyle interventions did not affect depression significantly.

Figure 2 shows the meta-analysis of lifestyle interventions on depression in women. In this case, lifestyle interventions led to a reduction in depression (Hedges’s g = −0.27; 95% CI −0.39, −0.14; Z = −4.17; *p* < 0.001; *I*^2^ = 75.25%). Owing to the insufficient number of studies, a similar meta-analysis for the male population could not be procured.

Table 3 shows the meta-analysis of lifestyle interventions on depression based on depression scales. Lifestyle interventions on depression in the Beck Depression Inventory (BDI) indicated a reduction in depression post-intervention [Hedges’ g = −0.26; 95% CI −0.45, −0.07; Z = −2.62; *p* = 0.009; *I*^2^ = 73.07%). Lifestyle interventions on depression in the Center for Epidemiologic Studies Depression Scale (CES-D) showed that lifestyle interventions led to a reduction in depression [Hedges’ g = −0.23; 95% CI −0.32, −0.14 (Z = −4.97; *p* < 0.001; *I*^2^ = 49.69%). Lifestyle interventions on depression in the Hospital Anxiety and Depression Scale (HADS) showed that lifestyle interventions led to a reduction in depression [Hedges’ g = −0.25; 95% CI −0.35, −0.14; Z = −4.62; *p* < 0.001; *I*^2^ = 30.95%). Lifestyle interventions on depression in the Patient Health Questionnaire (PHQ) showed that lifestyle interventions led to a reduction in depression [Hedges’s g = −0.16; 95% CI −0.28, −0.05 (Z = −2.76; *p* = 0.006; *I*^2^ = 49.46%).

### 3.5. Lifestyle Intervention and Anxiety

A meta-analysis of 47 randomized clinical trials of lifestyle interventions on anxiety showed that lifestyle interventions led to a reduction in anxiety, according to which Hedges’s g was equal to −0.24 with a 95% confidence interval of −0.32, −0.15 (Z = −5.54; *p* < 0.001; *I*^2^ = 59.25) (Figure 3).

### 3.6. Sub-Group Analysis Lifestyle Intervention and Anxiety

Figure 4 shows the meta-analysis of lifestyle interventions for anxiety based on different populations. Lifestyle interventions on anxiety in the cancer population showed that they led to a reduction in anxiety (Hedges’s g = −0.32; 95% CI −0.58, −0.06; Z = −2.42; *p* = 0.015; *I*^2^ = 47.10%). Lifestyle interventions for anxiety in the heart-related disease population showed that lifestyle interventions led to a reduction in anxiety [Hedges’s g = −0.26; 95% CI −0.43, −0.10; Z = −3.17; *p* = 0.002; *I*^2^ = 30.52%]. Lifestyle interventions on anxiety in other mental disorder populations showed that they led to a reduction in anxiety (Hedges’s g = −0.35; 95% CI −0.64, −0.07; Z = −2.46; *p* = 0.014; *I*^2^ = 0%). Lifestyle interventions on anxiety in the stroke population reduced anxiety (Hedges’s g = −0.29; 95% CI −0.52, −0.06; Z = −2.42; *p* = 0.015; *I*^2^ = 0%). Lifestyle interventions for anxiety in depressed, diabetic, overweight/obese, and obstructive pulmonary disease populations showed that their effects on anxiety were not significant.

Figure 5 shows the meta-analysis of lifestyle interventions for anxiety among women. Lifestyle interventions for anxiety in women led to reduced anxiety (Hedges’ g = −0.29; 95% CI −0.47, −0.10; Z = −3.04; *p* = 0.002; *I*^2^ = 72.86%). Owing to the insufficient number of studies, a similar meta-analysis for the male population could not be accomplished.

Figure 6 shows a meta-analysis of lifestyle interventions for anxiety based on anxiety scales. Lifestyle interventions on anxiety in the Brief Symptom Inventory (BSI) showed that lifestyle interventions led to a reduction in anxiety (Hedges’s g = −0.27; 95% CI −0.48, −0.06; Z = −2.52; *p* = 0.012; *I*^2^ = 0%). Lifestyle interventions for anxiety in the DASS showed a reduction in anxiety [Hedges’ g = −0.23; 95% CI −0.42, −0.05 (Z = −2.46; *p* = 0.014; *I*^2^ = 59.86%). Lifestyle interventions on anxiety in Generalized anxiety disorder (GAD) showed a reduction in anxiety (Hedges’s g = −0.47; 95% CI −0.76, −0.18; Z = −3.15; *p* = 0.000; I2 = 43.36%). Lifestyle interventions for anxiety in the HADS showed a reduction in anxiety (Hedges’ g = −0.25; 95% CI −0.34, −0.15; Z = −5.13; *p* = 0.001; *I*^2^ = 8.37%). Lifestyle interventions on anxiety in the SCL−90 showed a reduction in anxiety (Hedges’s g = −0.42; 95% CI −0.77, −0.07; Z = −2.34; *p* = 0.019; *I*^2^ = 0%). Lifestyle interventions for anxiety were not significant in the STAI group.

### 3.7. Lifestyle Intervention and Stress

A meta-analysis of 27 randomized clinical trials of lifestyle interventions on stress showed a reduction in stress, according to which Hedges’ g was equal to −0.22 with a 95% confidence interval −0.34, −0.11 (Z = −3.80; *p* < 0.001; *I*^2^ = 61.40) (Figure 7).

Figure 8 shows a meta-analysis of lifestyle interventions on stress based on different populations. Lifestyle interventions for stress in depressed populations showed a reduction in stress [Hedges’ g = −0.63; 95% CI −0.96, −0.31; Z = −3.79; *p* < 0.001; *I*^2^ = 0%). Lifestyle interventions for stress in the heart-related disease population showed a reduction in stress [Hedges’s g = −0.41; 95% CI −0.64, −0.18; Z = −3.50; *p* < 0.001; *I*^2^ = 0%). Lifestyle interventions for stress in cancer, diabetes, and overweight/obese populations showed that the effects of lifestyle interventions on stress were not significant.

Figure 9 shows a meta-analysis of lifestyle interventions for stress in women. Lifestyle interventions on stress in women showed a reduction in stress [Hedges’ g = −0.20; 95% CI −0.37, −0.03 (Z = −2.25; *p* = 0.024; *I*^2^ = 64.27%). Owing to the insufficient number of studies, a similar meta-analysis on the male population could not be performed.

Figure 10 shows the meta-analysis of lifestyle interventions for stress based on the stress scales. Lifestyle interventions for stress in the DASS showed a reduction in stress [Hedges’ g = −0.31; 95% CI −0.51, −0.10; Z = −2.96 2; *p* = 0.003; *I*^2^ = 59.42%). Lifestyle interventions on stress in the Perceived Stress Scale (PSS) showed a reduction in stress (Hedges’ g = −0.17; 95% CI −0.31, −0.03; Z = −2.45; *p* = 0.014; *I*^2^ = 60.77%]).

### 3.8. Publication Bias and Heterogeneity

In a meta-analysis of lifestyle interventions on depression, the Q test showed 202.62 (d.f. 88; *p* < 0.001), and *I*^2^ was 56.57%, and showed moderate heterogeneity [59]. The funnel plot in Figure 11 showed that there is a publication bias. Egger’s test indicated *p* < 0.001 and showed publication bias. The trim-and-fill imputed 14 studies, and the adjusted Hedges’ g was equal to −0.14 with 95% confidence intervals −0.20, −0.08 [64].

In a meta-analysis of lifestyle interventions on anxiety, the Q test showed 112.89 (d.f 47; *p* < 0.001), and *I*^2^ was 59.25%, and showed moderate heterogeneity [59]. The funnel plot in Figure 12 indicates the publication bias. Egger’s test was *p* = 0.002 and showed publication bias; the trim-and-fill imputed four studies, and Hedges’ g was equal to −0.20 with a 95% confidence interval of −0.29, −0.12 [64].

In a meta-analysis of lifestyle interventions on stress, the Q test showed 67.27 (d.f 26; *p* < 0.001), and *I*^2^ was 61.40%, and showed substantial heterogeneity [59]. The funnel plot in Figure 13 shows no publication bias. The Egger’s test scored *p* = 0.139 and did not show publication bias; the trim-and-fill test [64] did not impute any study.

## 4. Discussion

This systematic review and meta-analysis investigated the effects of lifestyle interventions on depression, anxiety, and stress in randomized clinical trials. This study included 96 eligible clinical trials to address research gaps noted in previous meta-analyses.

The results showed that lifestyle interventions led to improvements in depression, anxiety, and stress levels. This means that as people adopt a healthy lifestyle, their mental health improves. The finding related to the effect of lifestyle intervention on depression and anxiety is consistent with studies that have shown this effect [39,40], with the difference that the scope of the current study was much wider, and it was also methodologically strong because previous meta-analysis studies sometimes included clinical trials without a control group, or they combined an individual randomized clinical trial with a cluster. They also used non-parametric statistics, which can reduce the accuracy of the results, and all these factors can lead to weakness. A previous meta-analysis also showed a large effect size for the effect of lifestyle interventions on anxiety; however, that study was limited by the small number of studies included in the meta-analysis and the study population of overweight and obese women [161]. Unlike these previous analyses, our study systematically reviewed and meta-analyzed the impact of stress, which is a novel contribution to this field.

Our findings revealed that lifestyle interventions significantly reduced stress, with pronounced effects in individuals with depression, heart disease, and in women. The considerable role of stress in overall health has driven researchers to explore stress reduction methods over the past decade [162,163,164]. The results showed that lifestyle changes, such as exercise, diet, and improved sleep quality, can effectively reduce stress by lowering cortisol and increasing endorphin levels. Furthermore, the findings suggest that psychological factors, such as increased mindfulness and interoceptive awareness, may mediate these benefits [162,165]. Furthermore, stress and sleep quality are interrelated, and each affects the other in a bidirectional manner [166,167]. Moreover, sleep quality affects stress, and is also affected by stress, forming a vicious loop [168]. Similarly, while a healthy diet seems to reduce stress levels, higher levels of stress have been found to negatively impact diet quality [169]. In such cases, where a causes b, but also b causes a, it is important to target elements of the cycle that can be easier to break, which in such cases may be lifestyle changes rather than stress reductions. Additionally, among the three variables explored in this study (stress, depression, and anxiety), the effect size for lifestyle interventions was the highest for stress, suggesting a more pronounced effect. This further indicates the significance of the findings presented in this study, as stress has a direct impact on human health and influences epigenetic regulation [170]. Despite these negative effects, greater public awareness is required to highlight these direct links [171].

While previous reviews analyzing lifestyle interventions and depression have reported small and moderate effect sizes [43,172], the current review adds to the literature by confirming a modest effect size. In the current study, the effect of lifestyle interventions on depression was significant for individuals with depression, heart-related diseases, diabetes/at-risk diabetes, cancer, and metabolic syndrome, and for women. Interventions, such as healthy eating, increased physical activity, and exercise, have been found to have positive effects. A recent systematic review concluded that even low amounts of physical activity in a week can reduce the risk of developing depression by up to 18% compared to no activity [173].

Interventions based on a healthy lifestyle can affect mental health and reduce depression, anxiety, and stress through several mechanisms. One mechanism for the impact of lifestyle interventions on depression, anxiety, and stress involves neural mechanisms [174]. Physiological factors mediating the effects of physical activity and depression have been well studied, with findings that the effects of physical activity (as a lifestyle component) and antidepressant drugs on the relief of depression can occur through common neuro-molecular mechanisms [175,176] by increasing serotonin and norepinephrine, regulating the hypothalamus–pituitary–adrenal axis, and reducing systemic inflammatory signaling [177,178,179,180]. For anxiety and stress relief, studies have also shown similar neural mechanisms [181,182,183]. which helps reduce anxiety and stress. Healthy nutrition is another lifestyle mechanism that improves mental health [184]. For example, eating foods rich in carbohydrates can lead to diabetes and obesity [185] and. as studies have widely shown, obesity and diabetes are two important risk factors for depression, anxiety, and stress and lead to the deterioration of mental health [24,25,186,187,188,189]. Physiological mediating factors have also been explored to understand the role of a healthy diet in depression and overall affect, with some indications that the microbiome–gut–brain axis may be at its heart [187].

Anxiety: Similar to the findings regarding stress and depression, this study also found that physical activity, nutrition, and psychoeducation improved anxiety. The association between lifestyle interventions and reduced anxiety was prominent among patients with cancer, heart-related diseases, mental disorders, and women. With regard to the effect of physical activity on anxiety and stress relief, studies have also shown similar neural mechanisms [170,171,172], which help reduce anxiety and stress, as reported above. A healthy diet can positively affect anxiety through various mechanisms. These include the role of antioxidants, omega-3 fatty acids, zinc, probiotics, magnesium, and selenium in reducing the symptoms of anxiety disorders (citation). In the case of insufficient antioxidants, for example, oxidative stress has been linked to anxiety through pathways such as alterations in neurotransmission and neuronal function (citation). Moreover, an unhealthy diet can cause depression and anxiety by increasing blood glucose and glycemic load. It has been shown in animal studies that this concentration of high dietary glycemic load “leads to a decrease in plasma glucose to concentrations that trigger the secretion of autonomic counter-regulatory hormones, such as cortisol, adrenaline, growth hormone, and glucagon” [173,174,179]. Therefore, the effectiveness of lifestyle interventions on mental health based on the intensity and type of lifestyle can differ. In the studies included in this meta-analysis, there were differences in the lifestyle methods used, which affected the results of each study. Compared to other mental health interventions, lifestyle-based interventions may not be effective alone in improving mental health problems. Moreover, the effects of lifestyle interventions may not be achieved quickly, and therefore, other treatments, such as psychological and medicinal, also need to be considered. The effectiveness of lifestyle interventions on mental health is known [190], but how the costs and other aspects of this type of intervention compared to other psychological and pharmaceutical treatments compares need comparative study in the future.

Another significant finding was the effect of lifestyle interventions on depression, anxiety, and stress in women, confirming improvements across all three mental health domains. This study also revealed that the effectiveness of lifestyle interventions varied according to the scales used for assessment, with some yielding more significant results than others. Furthermore, the outcomes differed according to the patient population. For instance, depression showed a greater improvement among patients with metabolic disorders, or depression and cancer, whereas anxiety improved the most among those with depression and heart disease. These findings advocate lifestyle interventions as a component of comprehensive mental health care and highlight the need for public education on the connection between lifestyle and mental well-being.

### Strengths and Limitations

This study comprehensively reviewed common mental disorders, such as depression, anxiety, and stress, simultaneously in a systematic review and meta-analysis. In previous meta-analyses, different populations were not investigated. However, this distinction was made in this meta-analysis. This is because each population suffers from different diseases that can alter the effects of lifestyle interventions. Investigating gender differences was the study’s focus, and it was able to report results based on women separately; however, owing to the lack of studies, this review could not be performed for men. In addition, this study examined depression, anxiety, and stress based on different scales, which are the most important strengths of this meta-analysis. There are some limitations. These studies have primarily examined the effect of lifestyle interventions on depression and anxiety symptoms but not on depression and anxiety disorders, except for a few cases. Therefore, the generalization of the results to depression, anxiety, and stress disorders is limited. Each clinical trial on lifestyle has used different protocols and, although they have several commonalities, this heterogeneity might also impact the results. Variability in intervention types, such as the nature and intensity of lifestyle modifications, may affect the comparability of results across studies. Furthermore, the use of diverse measurement scales for mental health outcomes, although necessary for comprehensive analysis, introduces potential inconsistencies. Future studies could benefit from standardizing intervention protocols and measurement tools to enhance the comparability and robustness of their findings. Future studies should investigate the long-term impact of lifestyle interventions on mental health outcomes with an emphasis on their influence across broader demographic groups. Furthermore, analyzing subgroups, such as persons with diverse baseline mental health severities or differing socio-economic statuses, could provide more profound insights into the effectiveness and scalability of lifestyle interventions.

## 5. Conclusions

The findings showed the extent of the effectiveness of lifestyle-based interventions in improving mental health conditions, involving depression, anxiety, and stress. In addition, compared to other psychological and drug treatments, this type of intervention can be less expensive, healthier, and can be performed by more people. Therefore, considering and emphasizing these types of interventions can be highly beneficial and may have a long-term impact.

## Figures and Tables

**Figure 1 healthcare-12-02263-f001:**
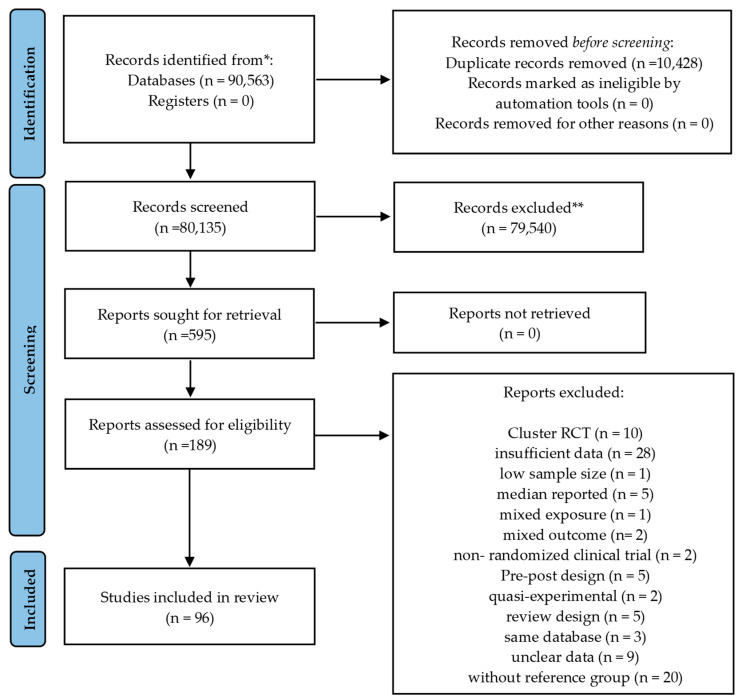
Flowchart diagram of screening studies included in this meta-analysis [51]. * Consider, if feasible to do so, reporting the number of records identified from each database or register searched (rather than the total number across all databases/registers). ** If automation tools were used, indicate how m7 any records were excluded by a human and how many were excluded by automation tools.

**Figure 2 healthcare-12-02263-f002:**
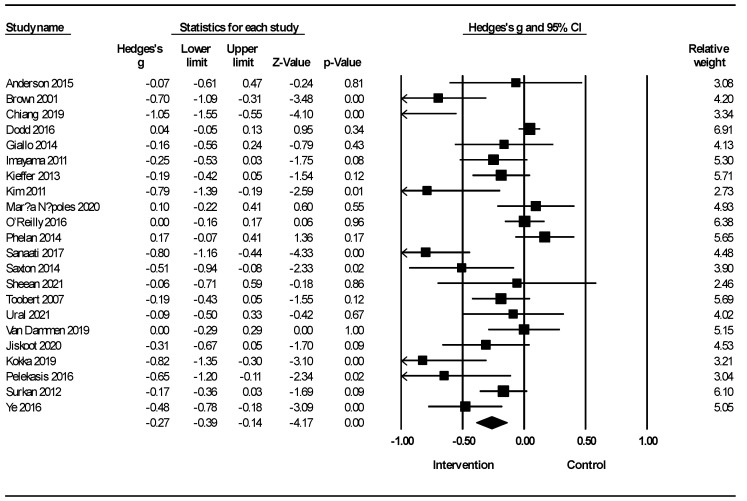
Forest plot for lifestyle intervention on depression in women [66,70,76,81,85,94,101,102,104,109,120,121,124,125,128,133,137,138,147,150,156,157].

**Figure 3 healthcare-12-02263-f003:**
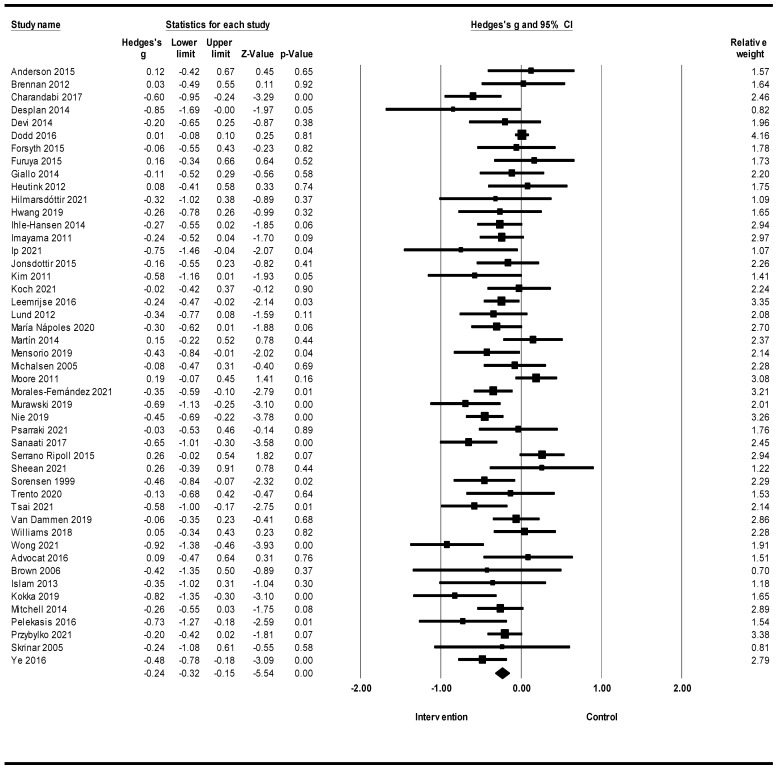
Forest plot of lifestyle intervention on anxiety [65,66,68,71,75,79,80,81,82,83,85,89,90,92,94,96,97,98,102,103,104,107,108,109,110,112,113,117,118,122,124,127,128,129,135,136,138,142,143,150,151,155,157,158,159,160].

**Figure 4 healthcare-12-02263-f004:**
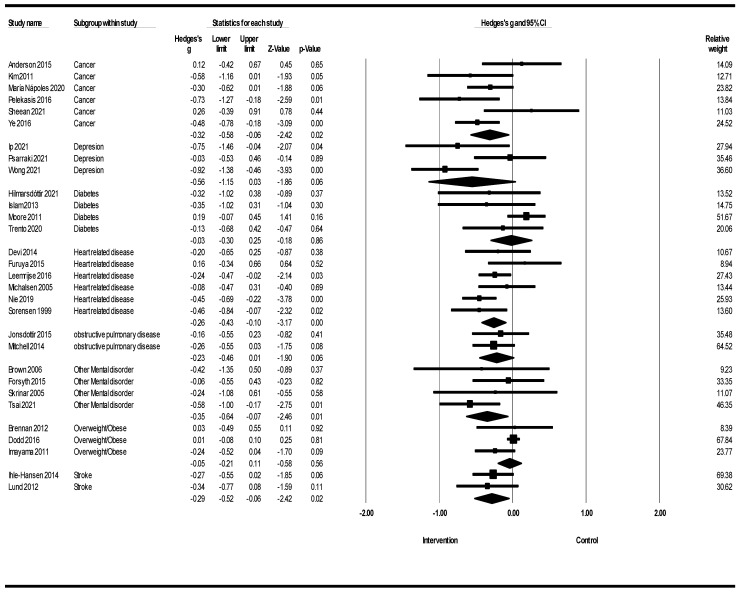
Forest plot for lifestyle intervention on anxiety based on diseases [66,68,71,80,81,82,83,90,93,94,96,97,98,102,107,108,109,113,118,122,128,129,135,136,143,150,155,157,158,160].

**Figure 5 healthcare-12-02263-f005:**
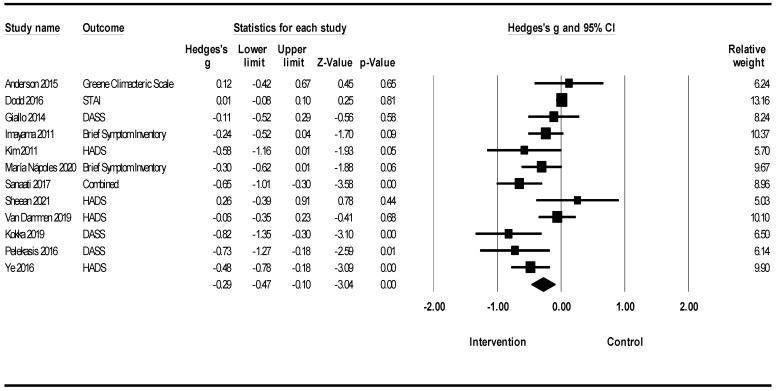
Forest plot for lifestyle intervention on anxiety in women [66,81,85,94,102,104,109,124,128,138,150,157].

**Figure 6 healthcare-12-02263-f006:**
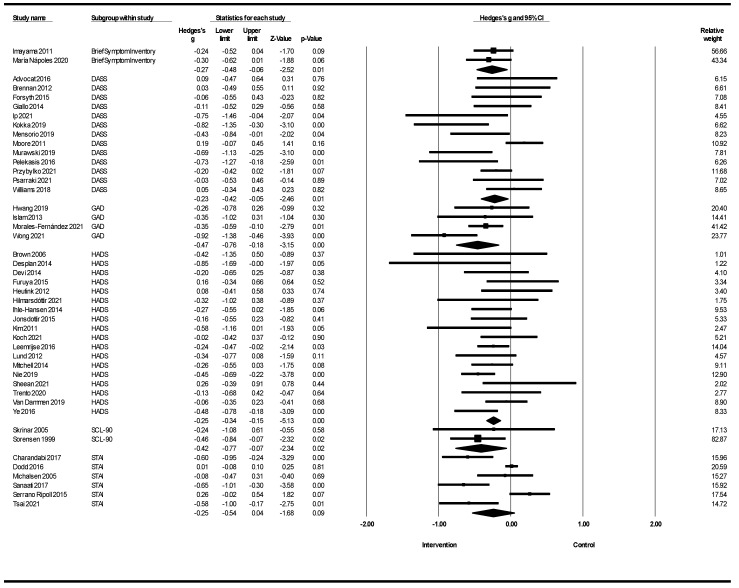
Forest plot for lifestyle intervention on anxiety based on anxiety scales [65,68,71,75,79,80,81,82,83,85,89,90,92,93,94,97,98,102,103,104,107,108,109,112,117,118,122,127,128,129,135,136,138,143,150,155,157,158,159,160].

**Figure 7 healthcare-12-02263-f007:**
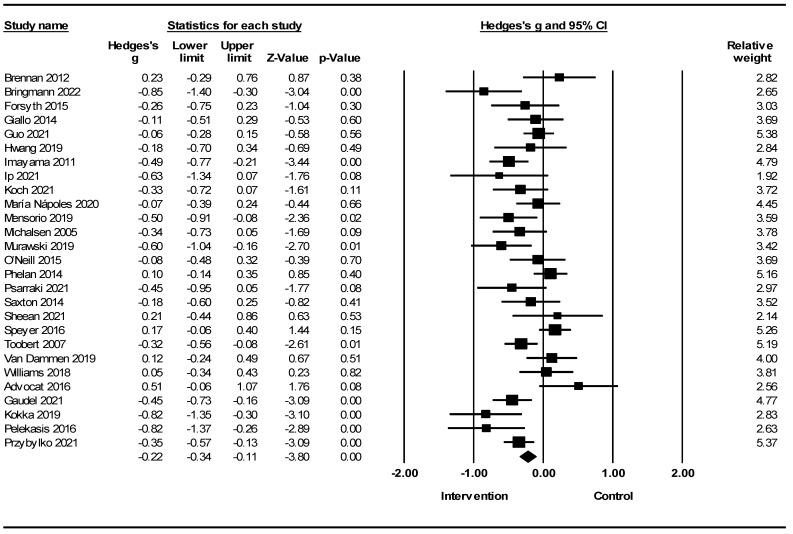
Forest plot for lifestyle intervention on stress [65,68,69,82,85,87,92,94,96,103,104,109,112,113,117,119,121,122,125,128,130,133,142,145,150,151,161].

**Figure 8 healthcare-12-02263-f008:**
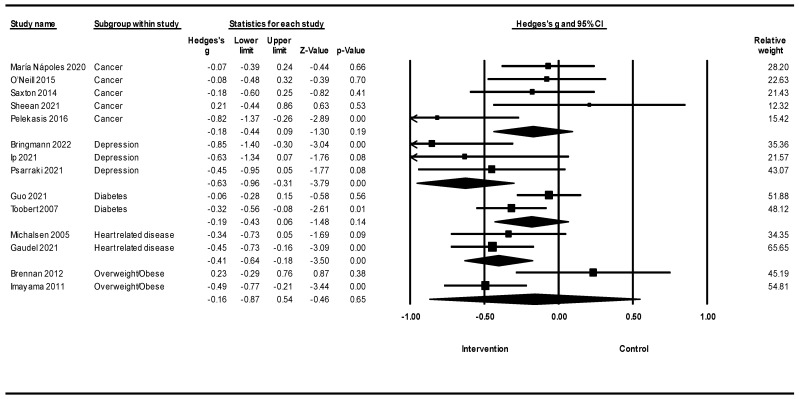
Forest plot for lifestyle intervention on stress based on diseases [68,69,82,94,96,113,119,122,125,128,133,145,150].

**Figure 9 healthcare-12-02263-f009:**
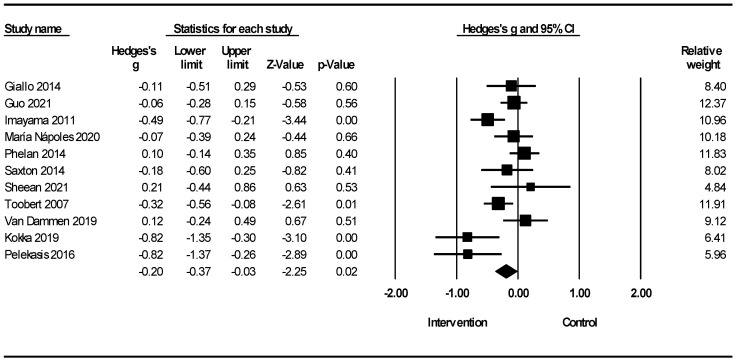
Forest plot for lifestyle intervention on stress in women [85,87,94,103,109,121,125,128,133,138,150].

**Figure 10 healthcare-12-02263-f010:**
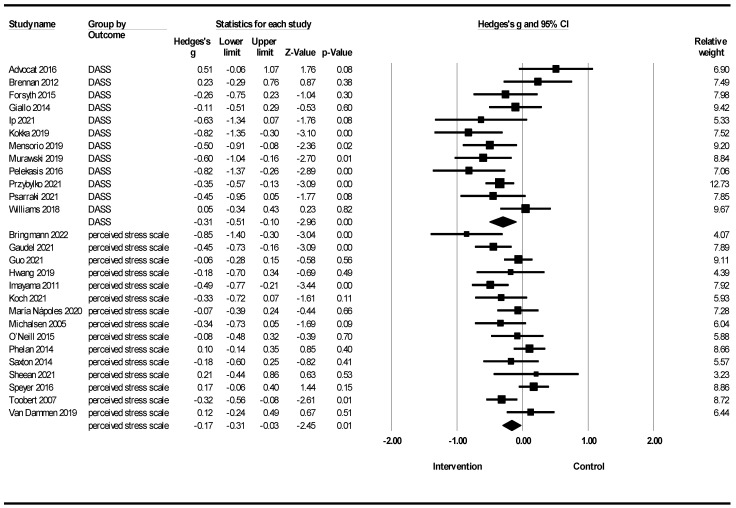
Forest plot for lifestyle intervention on stress based on stress scales [65,68,69,82,85,87,94,96,103,109,112,113,117,119,121,122,125,128,130,133,138,142,144,145,147,150,151].

**Figure 11 healthcare-12-02263-f011:**
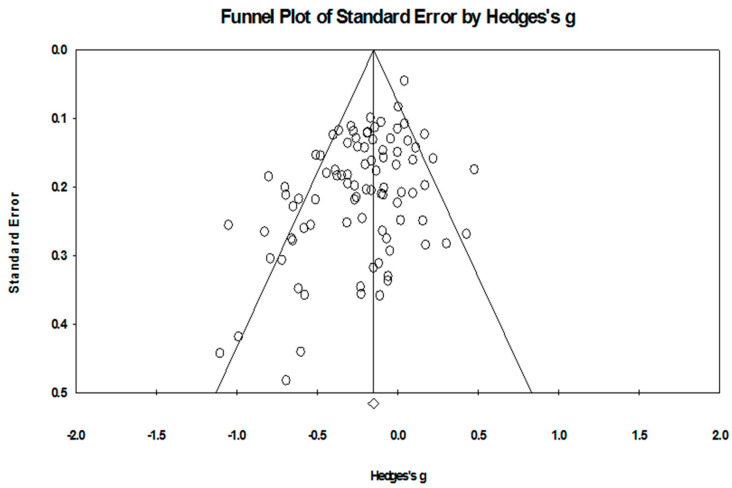
Funnel plot for lifestyle intervention and depression.

**Figure 12 healthcare-12-02263-f012:**
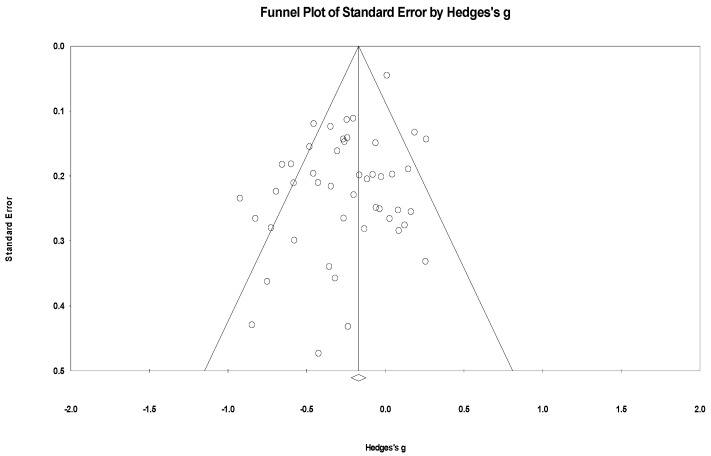
Funnel plot for lifestyle intervention and anxiety.

**Figure 13 healthcare-12-02263-f013:**
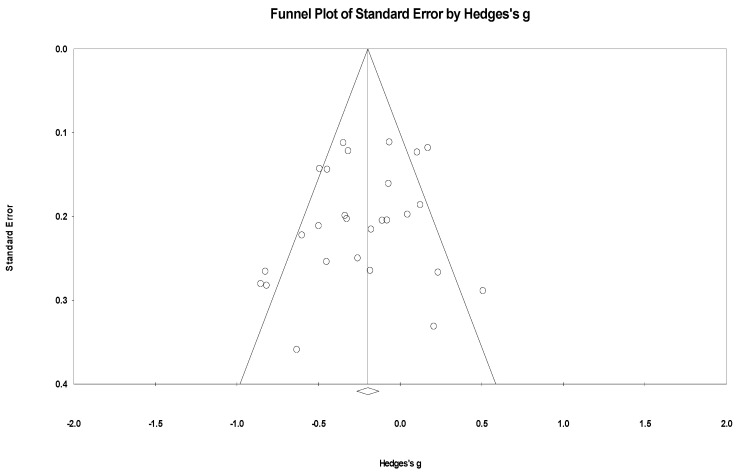
Funnel plot for lifestyle intervention and stress.

**Table 2 healthcare-12-02263-t002:** Lifestyle intervention on depression based on diseases.

Number of Studies	Disease	Hedges’s g	Lower Limit	Upper Limit	Z Value	*p*	*I* ^2^
**7**	Cancer	−0.34	−0.59	−0.08	−2.54	**0.011**	56.23%
10	Depression	−0.44	−0.62	−0.26	−4.82	**0.000**	40.46%
18	Diabetes/at risk of diabetes	−0.15	−0.27	−0.03	−2.43	**0.015**	56.51%
8	Heart-related disease	−0.19	−0.34	−0.04	−2.44	**0.015**	39.52%
6	Other mental disorders	−0.01	−0.17	0.15	−0.11	0.914	0%
2	Metabolic syndrome	−0.74	−1.27	−0.21	−2.76	**0.006**	69.66%
3	obstructive pulmonary disease	−0.14	−0.33	0.05	−1.44	0.151	0%
4	Older adults	−0.09	−0.23	0.05	−1.27	0.204	0%
4	Overweight/obesity	0.03	−0.19	0.24	0.25	0.802	53.18

**Table 3 healthcare-12-02263-t003:** Lifestyle intervention on depression based on depression scales.

Number of Studies	Scale	Hedges’s g	Lower Limit	Upper Limit	Z Value	*p*	*I* ^2^
15	Beck Depression Inventory	−0.26	−0.45	−0.07	−2.62	**0.009**	73.07%
12	Center for Epidemiologic Studies Depression Scale	−0.23	−0.32	−0.14	−4.97	**0.000**	12.53%
14	Depression anxiety and stress scale	−0.15	−0.31	0.02	−1.76	0.078	49.69
4	Edinburgh Postnatal Depression Scale	−0.23	−0.58	0.13	−1.23	0.217	89.06%
3	Geriatric Depression Scale	−0.31	−0.71	0.10	−1.49	0.136	60.71%
19	Hospital anxiety and depression scale	−0.25	−0.35	−0.14	−4.62	**0.000**	30.95%
16	Patient Health Questionnaire	−0.16	−0.28	−0.05	−2.76	**0.006**	49.46%

## Data Availability

Data sharing is not applicable. No new data were created or analyzed in this study.

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
