# Peer review of "The Effect of Lifestyle Interventions on Anxiety, Depression and Stress: A Systematic Review and Meta-Analysis of Randomized Clinical Trials"

_healthcare, 2024, doi:10.3390/healthcare12222263_

Round 1
Reviewer 1 Report
Comments and Suggestions for Authors
Given the substantial body of existing literature on this subject, including several papers with similar themes and titles, I feel that the proposed topic does not offer enough novelty to stand out in the current landscape of research. While lifestyle interventions remain an important area of mental health research, the lack of a fresh angle or new findings makes this particular manuscript less likely to contribute significant new insights to the field.
Author Response
Reveiwer 1 |
Given the substantial body of existing literature on this subject, including several papers with similar themes and titles, I feel that the proposed topic does not offer enough novelty to stand out in the current landscape of research. While lifestyle interventions remain an important area of mental health research, the lack of a fresh angle or new findings makes this manuscript less likely to contribute significant new insights to the field.
|
I appreciate your thoughtful feedback. We appreciate your concerns regarding originality and would like to express that our study offers new insights by focusing on several areas that are often overlooked. In contrast to many reviews, we prioritize stress as a key outcome in addition to depression and anxiety, providing a more thorough understanding of how lifestyle interventions affect mental health, especially in the context following COVID-19. Furthermore, we conduct subgroup analyses based on demographics, including age and gender, which offers a more detailed insight into how various groups respond to these interventions. Our review categorizes lifestyle changes into specific types, including physical activity and dietary improvements, to identify which are most effective for particular outcomes. Finally, we would like to highlight the importance of using diverse measurement tools, as they enhance our findings and facilitate more dependable comparisons across studies. We hold the view that these aspects set our work apart and contribute positively to the field. I appreciate the chance to provide clarification once again. |

Reviewer 2 Report
Comments and Suggestions for Authors
The manuscript “The Effect of Lifestyle Interventions on Common Mental Disorders: A Systematic Review and Meta-Analysis of Randomized Clinical Trials” by Amiri and colleagues is a systematic review and meta-analysis of randomized clinical trials evaluating the effects of lifestyle interventions on depressive and anxiety symptoms in humans diagnosed with a large body of medical conditions. The results obtained are interesting and important, methods were accurately reported and conducted following the world-accept guidelines for systematic reviews and meta-analysis but many flaws in results reporting and discussion overshadow the manuscript’s strengths, and it is not acceptable for publication in the Journal Healthcare in the present form. Please find above the detailed comments that should be considered by the authors before I can endorse the manuscript publication:
1. The title gives the wrong feeling that the study evaluated the effect of lifestyle interventions on mental disorders. Instead, the study assessed symptoms related to depressive and anxiety disorders, as stated by themselves. I suggest a title alteration to better cover the aspects of the systematic review.
2. The abstract would benefit from improvement, especially regarding the results and discussion/conclusion sections. Please clearly state the type of lifestyle interventions (at least a summary of which is considered lifestyle intervention) and how they affected the outcomes. Moreover, it should be clear that the study reviewed depression and anxiety symptoms, instead of mental disorders.
3. The results section would benefit from improvement. The authors could use this section to state better their results, for example, how many studies were selected for depression, and how many were used for anxiety, making it easier to compare the number of studies in each case. Moreover, the authors provided a forest plot for subgroup analysis in anxiety and stress studies, but not for depression. This should be clarified.
4. An overall view of the quality of the studies assessed in the study should be provided, especially because the table is very long. A statement in this sense would improve the interpretation of the meta-analysis.
5. The discussion section would benefit from improvement. The results were discussed in face of the previously published meta-analysis, but the implication of each new result, especially regarding the new sub-group analysis, is missing. For example, the authors found that depression symptoms were improved when lifestyle interventions were applied to patients diagnosed with cancer and mental disorders, for example. The discussion should explore such differences. The results regarding stress symptoms are barely cited in the discussion section.
Minor:
1. Figure 1: the meaning of the asterisks presented in the figure was not explained.
Comments on the Quality of English LanguageAn extensive English revision must be performed. There are many problems in punctuation, phrase construction, and some sentences are repeated over the text (for example, please refer to lines 373-374 page 13).
Author Response
Reviewer 2 |
The manuscript “The Effect of Lifestyle Interventions on Common Mental Disorders: A Systematic Review and Meta-Analysis of Randomized Clinical Trials” by Amiri and colleagues is a systematic review and meta-analysis of randomized clinical trials evaluating the effects of lifestyle interventions on depressive and anxiety symptoms in humans diagnosed with a large body of medical conditions. The results obtained are interesting and important, methods were accurately reported and conducted following the world-accept guidelines for systematic reviews and meta-analysis but many flaws in results reporting and discussion overshadow the manuscript’s strengths, and it is not acceptable for publication in the Journal Healthcare in the present form. Please find above the detailed comments that should be considered by the authors before I can endorse the manuscript publication:
1. The title gives the wrong feeling that the study evaluated the effect of lifestyle interventions on mental disorders. Instead, the study assessed symptoms related to depressive and anxiety disorders, as stated by themselves. I suggest a title alteration to better cover the aspects of the systematic review.
2. The abstract would benefit from improvement, especially regarding the results and discussion/conclusion sections. Please clearly state the type of lifestyle interventions (at least a summary of which is considered lifestyle intervention) and how they affected the outcomes. Moreover, it should be clear that the study reviewed depression and anxiety symptoms, instead of mental disorders.
3. The results section would benefit from improvement. The authors could use this section to state better their results, for example, how many studies were selected for depression, and how many were used for anxiety, making it easier to compare the number of studies in each case. Moreover, the authors provided a forest plot for subgroup analysis in anxiety and stress studies, but not for depression. This should be clarified.
4. An overall view of the quality of the studies assessed in the study should be provided, especially because the table is very long. A statement in this sense would improve the interpretation of the meta-analysis.
5. The discussion section would benefit from improvement. The results were discussed in face of the previously published meta-analysis, but the implication of each new result, especially regarding the new sub-group analysis, is missing. For example, the authors found that depression symptoms were improved when lifestyle interventions were applied to patients diagnosed with cancer and mental disorders, for example. The discussion should explore such differences. The results regarding stress symptoms are barely cited in the discussion section.
Minor:
1. Figure 1: the meaning of the asterisks presented in the figure was not explained.
Comments on the Quality of English Language An extensive English revision must be performed. There are many problems in punctuation, phrase construction, and some sentences are repeated over the text (for example, please refer to lines 373-374 page 13). |
1. Thank you for the feedback. We have revised the title to reflect this distinction more accurately. The updated title is now “The Effect of Lifestyle Interventions on Depression, Anxiety, and Stress: A Systematic Review and Meta-Analysis of Randomized Clinical Trials.” This change aims to set appropriate expectations from the outset and align the title more closely with the study’s focus.
2. Thank you for your valuable feedback. In response, we have revised the abstract to clearly specify the types of lifestyle interventions such as physical activity, dietary changes, and sleep hygiene and clarified that our study focuses on symptoms of depression and anxiety rather than on mental disorder diagnoses.
3. The number of studies for each outcome was previously entered and highlighted ,89 (depression), 47 9(anxiety), 27 (stress). Because the number of depression studies was 89 and could not be included in Figure.
4. Thank you for your insightful suggestion. We have now included an overall quality assessment statement in Section 2.7: Study Risk of Bias Assessment to provide a clearer view of the quality of the studies assessed.
5. Thank you for your constructive feedback. We have expanded the discussion to better address the subgroup analyses, highlighting specific improvements in depression symptoms for patients with cancer and mental disorders, as well as among women. Also, we now provide a more detailed exploration of the results concerning stress reduction, emphasizing the significant impact of lifestyle interventions in mitigating stress across diverse patient groups. Minor
1. Descriptions were provided.
Thanks for the comments. This manuscript has been edited by the professional editors from Editage for English language grammar and accuracy. Please find attached certificate. |

Reviewer 3 Report
Comments and Suggestions for Authors
1-)correct the following sentence:
Methods: Five databases were systematically searched. including PubMed, Web of Science, Scopus, the Cochrane Library, and Google 18
Scholar.
2-)you can improve the sentence by improving the clarity of the sentence.
Compared to other psychological and drug treatments, this type of 28
intervention can be less expensive and can be done by more people.
3-)you can highlight novelty of study in the abstract.
4-)you can add following sentence to improve clarity.
The COVID-19 pandemic has impacted not just health but also social structures (Uludag, 2022).
Another mental condition that has increased especially since the 45
Covid-19 pandemic is stress [4].
https://www.ncbi.nlm.nih.gov/pmc/articles/PMC9026955/
Uludag, K. (2022). ’Coronary Blindness: Desensitization after excessive exposure to coronavirus-related information ‘. Health Policy and Technology, 11(3), 100625.
5-) you can give more information about the unclear data. how to define it?you may mention it in the methods section.
6-)you may improve the figure quality. it seems not very clear in the pdf file.
7-) you can mention overall results in the first paragraph of the discussion section.
8-) you can add suggestions for further studies.
9-) you can mention limitation of your studies.
You can also mention substance use cessation as a lifestyle intervention as in the previous study .
Opie, R.S.; Jacka, F.N.; Marx, W.; Rocks, T.; Young, C.; O’Neil, A. Designing Lifestyle Interventions for Common Mental Disorders: What Can We Learn from Diabetes Prevention Programs? Nutrients 2021, 13, 3766. https://doi.org/10.3390/nu13113766
Comments on the Quality of English Languagethe manuscript can be written more concisely.
Author Response
Reviewer 3 |
-)correct the following sentence: Methods: Five databases were systematically searched. including PubMed, Web of Science, Scopus, the Cochrane Library, and Google 18 2-)you can improve the sentence by improving the clarity of the sentence. Compared to other psychological and drug treatments, this type of 28 3-)you can highlight novelty of study in the abstract. 4-)you can add following sentence to improve clarity. The COVID-19 pandemic has impacted not just health but also social structures (Uludag, 2022). Another mental condition that has increased especially since the 45 https://www.ncbi.nlm.nih.gov/pmc/articles/PMC9026955/ Uludag, K. (2022). ’Coronary Blindness: Desensitization after excessive exposure to coronavirus-related information ‘. Health Policy and Technology, 11(3), 100625. 5-) you can give more information about the unclear data. how to define it?you may mention it in the methods section. 6-)you may improve the figure quality. it seems not very clear in the pdf file. 7-) you can mention overall results in the first paragraph of the discussion section. 8-) you can add suggestions for further studies. 9-) you can mention limitation of your studies.
You can also mention substance use cessation as a lifestyle intervention as in the previous study . Opie, R.S.; Jacka, F.N.; Marx, W.; Rocks, T.; Young, C.; O’Neil, A. Designing Lifestyle Interventions for Common Mental Disorders: What Can We Learn from Diabetes Prevention Programs? Nutrients 2021, 13, 3766. https://doi.org/10.3390/nu13113766.
Comments on the Quality of English Language the manuscript can be written more concisely.
|
Thank you for your thorough and insightful review. I appreciate the constructive feedback provided, which has helped enhance the clarity and quality of this manuscript
1. Updated the sentence to clearly describe the databases searched, emphasizing the use of Google Scholar for gray literature. 2. Revised the sentence comparing interventions to highlight cost-effectiveness and accessibility. 3. Added a description in the abstract to emphasize the study's unique focus on stress and its differential impacts across various populations. 4. Combined sentences on the pandemic's influence on mental health to enhance readability and coherence and appropriately referenced it. 5. Clarified how ambiguous data were categorized and addressed, following Cochrane quality assessment guidelines, and added details in the Methods section 2.7 Study risk of bias assessment 6. These are software outputs and cannot be changed. 7. An introductory summary of key results already have been added to enhance understanding of the overall findings. 8. We have suggested future research directions, such as examining long-term effects of lifestyle interventions across diverse demographics. 9. We have addressed limitations in the Discussion section, specifically regarding intervention variability and measurement scale differences. 10. This study doesn’t seem coherent with our aims and inclusion criteria.
Thanks for the comments. This manuscript has been edited by the professional editors from Editage for English language grammar and accuracy. Please find attached certificate. |

Reviewer 4 Report
Comments and Suggestions for Authors
The introduction presents a lot of relevant information about the prevalence of mental disorders and the factors that influence them. However, this section could benefit from a clearer structure that highlights the specific objectives of the study from the beginning. Additionally, it would be helpful if the introduction more briefly contextualized previous studies and quickly brought into focus the knowledge gaps that this article addresses.
Inclusion and exclusion criteria: Although the inclusion and exclusion criteria are mentioned in the methods section, they could be further detailed to avoid ambiguity. For example, the criteria on randomized controlled trials (RCTs) are clear, but it would be useful to justify why certain types of studies, such as quasi-experimentals, are excluded, or why studies with a mixed outcomes approach are not included.
Choice of databases: Although databases such as PubMed, Web of Science, Scopus and Cochrane Library were used, Google Scholar may not be the best option in terms of quality of indexed publications. It would be helpful to justify their inclusion and describe how potentially duplicate or low-quality studies that Google Scholar might suggest are handled.
Publication bias and heterogeneity: Heterogeneity (I2) and publication bias are reported, but how these factors affected the robustness of the results is not sufficiently explained. It would be important to discuss whether the detected release session has a significant impact on the conclusion of the meta-analysis and what corrective measures were taken (such as the use of release session correction methods such as Egger or funnel plots).
Although the discussion does a good job of presenting the results, it could be improved by offering a broader comparison with other studies and reviews. This would include a deeper analysis of why some lifestyle interventions were more effective than others, and whether there are specific characteristics (such as duration or intensity of the intervention) that influenced the observed effectiveness. Claims about the effectiveness of lifestyle interventions compared to other psychological or pharmacological treatments should be well supported with relevant citations. In this section, it would be useful to include comparative studies.
Comments on the Quality of English Language
While the English of the manuscript is generally good, there are some minor errors in writing and grammar that could be improved with careful proofreading. For example, some sentences in the introduction and summary may sound redundant or overloaded with information.
Author Response
Reviewer 4 |
1. The introduction presents a lot of relevant information about the prevalence of mental disorders and the factors that influence them. However, this section could benefit from a clearer structure that highlights the specific objectives of the study from the beginning. Additionally, it would be helpful if the introduction more briefly contextualized previous studies and quickly brought into focus the knowledge gaps that this article addresses. 2. Inclusion and exclusion criteria: Although the inclusion and exclusion criteria are mentioned in the methods section, they could be further detailed to avoid ambiguity. For example, the criteria on randomized controlled trials (RCTs) are clear, but it would be useful to justify why certain types of studies, such as quasi-experimentals, are excluded, or why studies with a mixed outcomes approach are not included. 3. Choice of databases: Although databases such as PubMed, Web of Science, Scopus and Cochrane Library were used, Google Scholar may not be the best option in terms of quality of indexed publications. It would be helpful to justify their inclusion and describe how potentially duplicate or low-quality studies that Google Scholar might suggest are handled. 4. Publication bias and heterogeneity: Heterogeneity (I2) and publication bias are reported, but how these factors affected the robustness of the results is not sufficiently explained. It would be important to discuss whether the detected release session has a significant impact on the conclusion of the meta-analysis and what corrective measures were taken (such as the use of release session correction methods such as Egger or funnel plots). 5. Although the discussion does a good job of presenting the results, it could be improved by offering a broader comparison with other studies and reviews. This would include a deeper analysis of why some lifestyle interventions were more effective than others, and whether there are specific characteristics (such as duration or intensity of the intervention) that influenced the observed effectiveness. Claims about the effectiveness of lifestyle interventions compared to other psychological or pharmacological treatments should be well supported with relevant citations. In this section, it would be useful to include comparative studies. 6. Comments on the Quality of English Language While the English of the manuscript is generally good, there are some minor errors in writing and grammar that could be improved with careful proofreading. For example, some sentences in the introduction and summary may sound redundant or overloaded with information.
|
1. Thank you for the feedback. We have reorganized the Introduction to clearly state the study’s objectives from the beginning. Additionally, we have provided context by discussing prior research on lifestyle interventions, which demonstrates their effectiveness in reducing symptoms of depression and anxiety. In lines 73-86, we have specifically highlighted gaps in the current literature, including the lack of focus on stress as a primary outcome. 2. Thank you for this observation. We have expanded the Inclusion and Exclusion Criteria section in the Methods to provide a clearer rationale for excluding certain study designs. 3. Thank you for the comments. To retrieve eligible articles, we systematically searched five databases: PubMed, Web of Science, Scopus, the Cochrane Library, and Google Scholar. Google Scholar was specifically used to identify gray literature, which may not be indexed in traditional databases but provides valuable insights into less-publicized research. We applied a rigorous screening process to exclude duplicate and potentially low-quality studies from Google Scholar, ensuring that only studies meeting quality criteria were included in the analysis. 4. The limitation caused by heterogeneity in this meta-analysis was included in the limitations section and we have expanded the strengths and limitations as per feedback. 5. Thank you for these insightful suggestions. In response, we have expanded the discussion to provide a broader comparison with existing studies and reviews, offering a deeper analysis of why certain lifestyle interventions appeared more effective. We have expanded the discussion with relevant citations as suggested.
6. Thanks for the comments. This manuscript has been edited by the professional editors from Editage for English language grammar and accuracy. Please find attached certificate. |
Round 2
Reviewer 1 Report
Comments and Suggestions for Authors
My concern was not about the originality of the study, rather it was about novelty. I hope the authors look for novel aspects in their coming studies. However, after revision, I think the paper is acceptable.
Reviewer 2 Report
Comments and Suggestions for Authors
The authors have adequately addressed my concerns, and I now consider the manuscript acceptable in its current form.